# Improving Complex SQL Generation for Text-to-SQL by Addressing Semantic Blind Spots in Pending SQL Components

## Abstract

In recent years, significant advancements in large language models (LLMs) have greatly propelled the development of Text-to-SQL tasks. However, due to the token-by-token sequential generation mechanism employed by these models, they encounter a `semantic blind spot` problem with respect to pending SQL components—the parts of the SQL query yet to be generated. Specifically, language models are unable to effectively utilize the semantic information of these pending SQL components during the generation of the final SQL query, which poses considerable challenges for generating complex SQL statements. To address this issue, we propose a novel thought process based on `SQL components pre-generation` and design a `maximum connected subtree matching` reward mechanism leveraging the SQL abstract syntax tree (AST) to improve the accuracy of local component generation. Extensive experiments demonstrate that, under comparable model parameter scales, our training approach achieves significant advantages, effectively enhancing the generation of complex SQL queries. Our method attains an execution accuracy EX of 65.78% on the BIRD-DEV dataset and achieves state-of-the-art (SOTA) performance on the Spider-syn datasets.

## 1 Introduction

The Text-to-SQL task aims to convert user natural language into SQL statements, enabling users without a deep understanding of SQL to query databases using natural language (Katsogiannis-Meimarakis & Koutrika, 2023). In recent years, significant progress has been made in this field with the application of LLMs technologies to the Text-to-SQL task (Liu et al., 2024). However, due to the token-by-token sequence generation mechanism of LLMs, they struggle to fully utilize the semantic information of the yet-to-be-generated parts of the SQL components, causing LLMs to still face challenges when generating complex SQL queries.

Specifically, our analysis shows that complex SQL statements are typically composed of multiple SQL components with independent semantics, such as SQL clauses and expressions. When humans write complex SQL queries, they do not necessarily do so in a strict sequential order. Instead, they may selectively construct certain SQL components first and then utilize the semantic information of these constructed components to gradually consider the remaining components and their interrelationships, ultimately forming the final SQL statement. For example, in nested SQL queries, it is common to write the inner SQL components first and then, based on their semantic information, design the outer SQL components step by step, thereby progressively completing the entire query.

However, due to the token-by-token generation mechanism of LLMs, the model can only produce the entire SQL statement sequentially. Unlike humans, it cannot selectively build certain SQL components first and then gradually handle the remaining components and their relationships. This causes most SQL components to be invisible to the model at the time of generating the final SQL, preventing the model from fully leveraging the semantic information of all components. We refer to this issue as the *semantic blind spots in pending SQL components*. As illustrated in Figure 1(a), when the model generates the final SQL, none of the SQL components have appeared in the already generated sequence, so all components are invisible to the model.

To address this problem, our preliminary idea is to control the reasoning process so that the model first generates the SQL components with independent semantic information before producing the final SQL statement. This way, when generating the final SQL, the model can have seen as many necessary SQL components as possible, thus alleviating the semantic blind spot problem to some extent. As shown in Figure 1(b), when generating the final SQL, two SQL components have already appeared in the generated sequence, making their semantics visible to the model. At this point, the model only needs to consider how to combine these components, thereby reducing the difficulty of generating the final SQL.

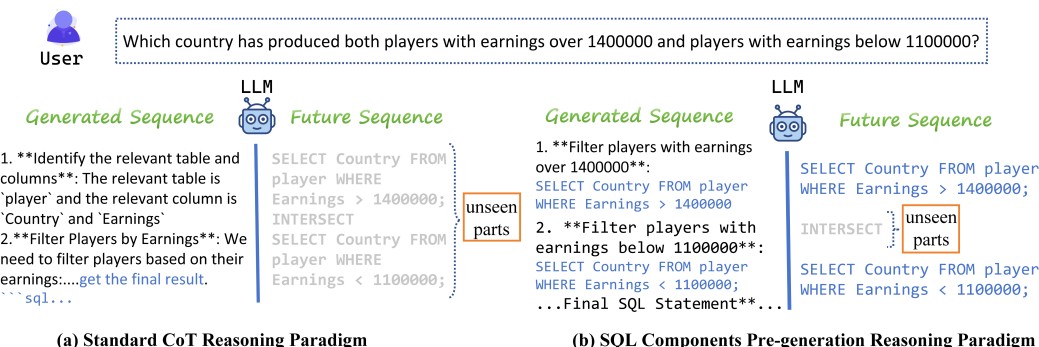

(a) Standard CoT Reasoning Paradigm          (b) SQL Components Pre-generation Reasoning Paradigm

Figure 1: Examples of Different Reasoning Paradigms. The left side represents the token sequence that has been generated so far, while the right side represents the token sequence yet to be generated.

Many existing Text-to-SQL works have focused on controlling the reasoning process to improve SQL generation performance. For example, studies such as (Dong et al., 2023; Pourreza & Rafiei, 2023; Pourreza et al., 2024) use Chain-of-Thought (CoT) methods (Wei et al., 2022; Zhou et al., 2022; Xie et al., 2024) to decompose tasks step-by-step during model reasoning, thereby enhancing SQL generation. There are also test-time scaling approaches that combine Monte Carlo Tree SearchCoulom (2006) to improve reasoning capabilities. With the emergence of DeepSeek-R1(Guo et al., 2025), the great potential of reinforcement learning has been recognized, and more works have begun to incorporate reinforcement learning to control the reasoning process, such as Reasoning-sql (Pourreza et al., 2025), Reward-sql (Zhang et al., 2025), and Rex-SQL (Dai et al., 2025). Although these methods significantly improve Text-to-SQL performance, they mainly optimize general reasoning processes and do not specifically consider the semantic independence of SQL components during generation. Therefore, the problem of semantic blind spots in pending SQL components remains unresolved.

To better optimize this problem, we focus on three main aspects for design and implementation.

*(1) When is SQL components pre-generation needed?* By analyzing SQL structures, we find that for simple SQL queries containing only basic elements (SQL keywords, column names, table names, etc.), SQL components pre-generation is meaningless. Therefore, we first estimate the complexity of the SQL to be generated. We borrow the SQL component-based complexity evaluation method from Spider(Yu et al., 2018) and only perform SQL components pre-generation for moderate and challenging SQL queries.

*(2) How to control the reasoning process for SQL components pre-generation?* Our experiments show that for weak models, it is difficult to control the reasoning process using only prompts. Thus, training the reasoning mode is still necessary. Considering the convenience of the group relative policy optimization (GRPO)(Shao et al., 2024) algorithm, we use GRPO to train the model's reasoning mode. During training, we design a reward function for SQL components pre-generation to guide the model's reasoning process as we expect.

*(3) How to improve the accuracy of local SQL components?* Errors in SQL components inevitably cause errors in the entire SQL statement. Therefore, improving SQL components accuracy is crucial. The n-gram matching method is a common way to measure local similarity. However, this sequential matching loses the structural information of SQL components, which can lead to incorrect rewards. For example, two different components "SELECT school, city" and "GROUP BY school, city" share the fragment "school, city". In 2-gram matching, this fragment is

mistakenly judged as a match, causing false rewards. To address this, we combine SQL AST and design a reward mechanism based on maximum connected subtree matching. This method effectively uses the structural information of SQL components for reward assignment. For the above example, the parent nodes of "`school, city`" in the two components are "`SELECT`" and "`GROUP BY`" respectively. Since the parent nodes do not match, the entire subtree matching fails, avoiding false rewards.

***In summary, our contributions are as follows:***

**(1)** We first identify the semantic blind spot in pending components and propose a complexity-guided SQL components pre-generation approach to solve it.

**(2)** We design a `maximum connected subtree matching reward` mechanism based on SQL AST to overcome the limitations of n-gram matching in local SQL component matching.

**(3)** We run experiments on BIRD and SPIDER datasets. Our method boosts complex SQL generation and outperforms similar-scale models on SPIDER variants.

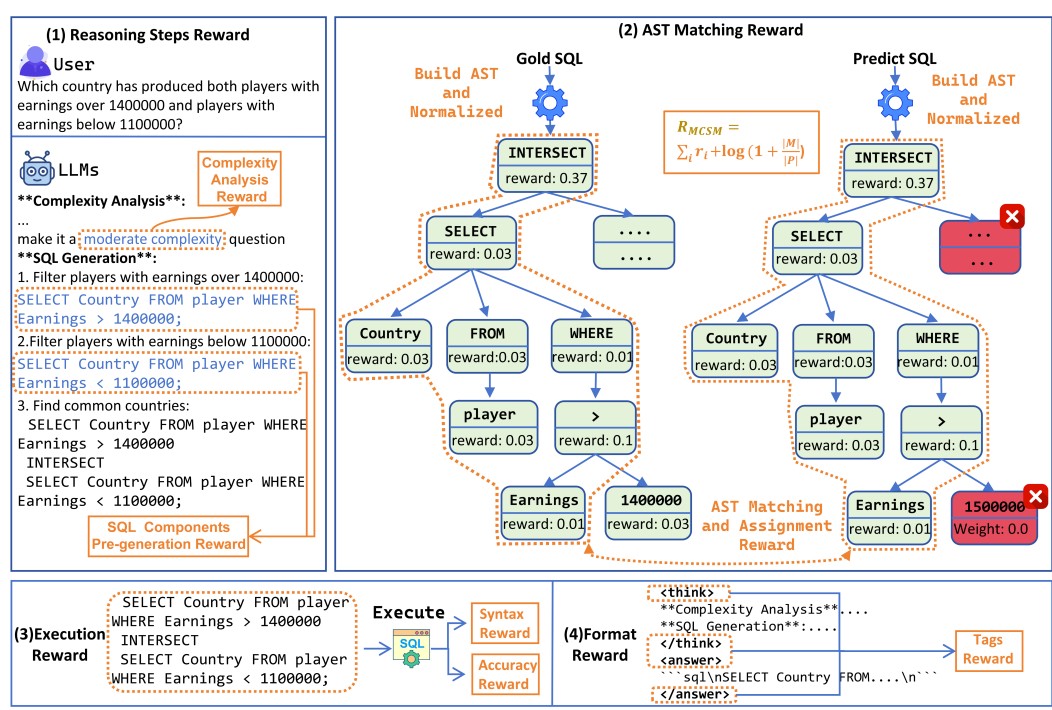

Figure 2: This diagram provides an overview of the four types of reward function designs. The orange-highlighted sections correspond to the respective reward function types.

## 2 METHODOLOGY

### 2.1 OVERVIEW

We train the model with the GRPO algorithm to align its reasoning process with our predefined design. To this end, we propose four reward types (Figure 2): (1) **Reasoning Steps Reward**: In this component, we design the reasoning steps to first analyze the complexity of the question. If it involves complex SQL, task decomposition is applied, and the corresponding SQL components are pre-generated simultaneously. Therefore, this reward includes a complexity analysis reward and a SQL Component Pre-generation Reward. (2) **AST Matching Reward**: To improve local SQL component accuracy, we design a maximum connected subtree matching reward. We build the gold SQL's AST $\mathcal{G}_{ast}$ with node rewards initialized by precomputed `inverse document frequency (IDF)` scores, then construct the predicted SQL's AST $\mathcal{P}_{ast}$. Both ASTs are normalized to enhance matching generality, and the reward is based on their comparison. (3) **Execution**

**Reward**: We assign a syntax reward if the SQL executes successfully, and an accuracy reward by comparing its execution result with the gold SQL. (4) **Format Reward**: Inspired by DeepSeek-r1, this reward encourages standardized output format, facilitating extraction of reasoning and answer components.

## 2.2 REASONING STEPS REWARD

As shown in Figure 2(1), the Reasoning Steps Reward includes: 1) Complexity Analysis Reward and 2) SQL Components Pre-generation Reward. Since simple SQL statements usually consist of a single table or column unit, we do not require pre-generation of SQL components. In these cases, we use the standard CoT reasoning and directly generate the final SQL after reasoning. Moderate and challenging SQL statements often differ in the number of components; for example, challenging SQL tends to have more nested queries and set operations. Thus, the reasoning steps reward is designed by first analyzing complexity and then setting the reasoning steps accordingly.

**(1) Complexity Analysis Reward:** We use the Spider dataset's method to classify SQL complexity into four levels: `easy`, `medium`, `hard`, and `extra hard`, based on SQL components. In the Bird training set, we label SQL accordingly—grouping hard and extra hard as `challenging`, easy as `simple`, and medium as `moderate`. During training, the model is also trained to predict complexity. If the prediction matches our label, it receives this reward, as shown in Figure 2(1) under `Complexity Analysis Reward`. Formulated as $\mathcal{R}_c = \mathbb{I}(\zeta(SQL) = \mathcal{M}(S, q))$, where $\zeta$ is the complexity evaluation function, $\mathbb{I}$ is the indicator function, $\mathcal{R}_c$ is the complexity analysis reward, and $\mathcal{M}(S, q)$ is the model's predicted complexity from schema $S$ and question $q$.

**(2) SQL Components Pre-generation Reward:** For `simple` cases, no SQL component pre-generation is needed. For `moderate` and `challenging` cases, the model must output at least two and three SQL components, respectively, within `<think>` and `</think>` tags to get this reward, as shown in Figure 2(1) under SQL Components Pre-generation Reward. It is defined as $R_p = \min\left(1.0, \frac{\|\mathbb{C}\|}{n}\right)$, where $R_p$ is the pre-generation reward, $\|\mathbb{C}\|$ is the number of generated components, and $n$ is the required minimum number of components.

## 2.3 AST MATCHING REWARD

To improve accuracy of local SQL components, we propose a maximum connected subtree matching reward. This reward allows partial matches since equivalent SQL queries can have multiple valid forms. We implement it in three steps:

**(1) Build AST:** We build ASTs for both gold SQL and predicted SQL. For the gold AST, each node is assigned a score to reflect the difficulty of correctly generating different keywords. For example, generating common keywords like `SELECT` and `FROM` is easier than generating complex keywords such as `INTERSECT`, which require consideration of logical relations between different SQL queries. To emphasize this, we use the IDF values of keywords computed from the training set as node scores. Since the values (e.g., constants) vary in each SQL, we assign the average IDF value of all nodes in the AST as the IDF for value nodes. After assigning scores to all nodes, we apply a softmax function to normalize the node scores, thus completing the gold AST construction and scoring. For the predicted AST, we only build its structure without assigning reward scores; scoring will be done later via matching with the gold AST.

**(2) AST Normalization:** Different SQL expressions with the same semantics often have different syntactic forms, making AST node matching difficult (see Figure 3(a)). For example, alias names in gold SQL might be `T1` and `T2`, whereas in predicted SQL they might be `E` and `V`. Also, the order of conditions in the `WHERE` clause, e.g., `votes >= 30` and `stars = 10`, might differ, causing mismatches (highlighted in red in the figure 3(a)). To reduce mismatches caused by such variations, we normalize both ASTs by removing alias nodes and sorting child nodes for order-insensitive keywords (like `AND`, `OR`, `=` etc.). After normalization, more nodes can be matched between gold and predicted ASTs, reducing false negatives. Figure 3(b) shows how normalization results in identical ASTs for both predicted and gold SQL.

**(3) AST Matching and Reward Assignment:** We traverse both gold and predicted ASTs simultaneously to find matched nodes, thereby identifying multiple maximum connected subtrees in the

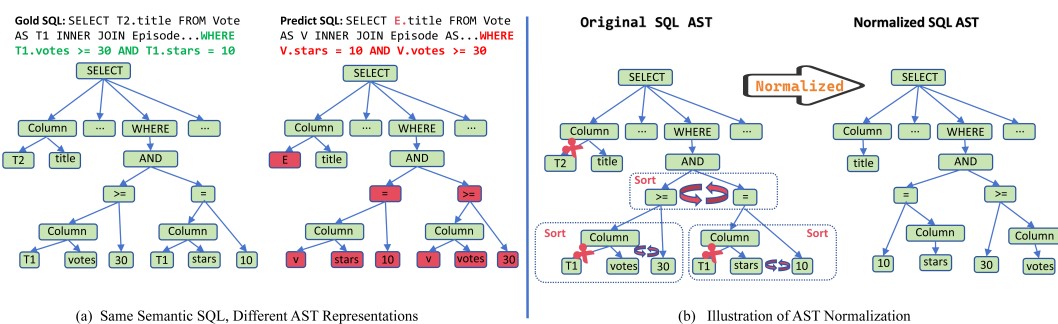

(a) Same Semantic SQL, Different AST Representations   (b) Illustration of AST Normalization

Figure 3: Impact of Alias and Order on AST Matching and Normalization Example

predicted AST. As shown in Figure 2(2), the orange dashed frame in the predicted AST marks a matched maximum connected subtree, while red nodes indicate unmatched nodes. In the illustration, only one maximum connected subtree is identified. The score of this subtree is computed as the reward for this portion, defined as $R_{\mathrm{MCSM}} = \sum_i r_i + \log\left(1 + \frac{|M|}{|P|}\right)$, where $R_{\mathrm{MCSM}}$ denotes the maximum connected subtree matching reward, $r_i$ is the reward assigned to the $i$-th matched node in the gold AST, $|M|$ is the number of nodes in the maximum connected subtree, and $|P|$ is the total number of nodes in the subtree containing this maximum connected subtree. For example, in Figure 2, the maximum connected subtree marked in the predicted AST (orange dashed area) contains 8 nodes ($|M| = 8$), representing the left subtree of an INTERSECT node which has 9 nodes in total ($|P| = 9$). This formulation encourages the model to match as many nodes as possible by increasing the reward proportional to the subtree size. Finally, the total AST matching reward over $K$ such maximum connected subtrees is calculated as $R_{\mathrm{ASTM}} = \sum_{k=1}^{K} R_{\mathrm{MCSM}}^{(k)}$.

### 2.4 BASE REWARDS

Base Rewards include Execution Reward (see Figure 2 (3)) and Format Reward (Figure 2 (4)). **(1) Execution Reward:** This consists of syntax and accuracy rewards. The predicted SQL is executed on the database; if no syntax errors occur, a syntax reward is given. If the execution result matches the gold SQL result, an accuracy reward is granted (Figure 2 (3)). **(2) Format Reward:** To aid extraction of reasoning and SQL parts, we adopt the format reward from DeepSeek-r1 (Guo et al., 2025). When the reasoning is enclosed in `<think>` tags and the predicted SQL in `<answer>` tags with correct pairing, the model receives this reward (Figure 2 (4)).

## 3 EXPERIMENTS

### 3.1 EXPERIMENTAL SETTINGS

**Datasets:** We conduct experiments on BIRD, SPIDER, and SPIDER 2.0Lei et al. (2024) datasets, which are popular benchmarks for Text-to-SQL tasks. For robustness evaluation, we use the SPIDER-SYN (Gan et al., 2021) and SPIDER-REAL (Deng et al., 2020) variants. Following Reasoning-SQL (Pourreza et al., 2025), we apply schema filtering from Chess (Talaei et al., 2024) during preprocessing. Importantly, only BIRD's training set is used for training; the others serve solely for validation. Detailed descriptions of these datasets are provided in the appendix A.1.

**Evaluation Metrics:** We use execution accuracy to evaluate SQL generation, VES to assess SQL execution efficiency, and exact match (EM) to measure improvements in SQL component accuracy. Detailed descriptions of these metrics are provided in the appendix A.2. For the `bird` dataset, we use the official execution accuracy script, and for the SPIDER dataset and its variants, we use the `spider-test-suite` script.

**Implementation Details:** Our experiments primarily use Qwen2.5-Coder-7B-Instruct (Hui et al., 2024). To verify the generality of our approach, we also evaluate on models of different scales, including the 3B and 14B versions of Qwen, as well as other models such as

Llama-3.1-8B-Instruct (Grattafiori et al., 2024) and deepseek-coder-6.7b-Instruct (Guo et al., 2024). The key experimental settings are as follows: `rollout_n=5`; reward weights include `reasoning_steps_reward=2` (Complexity analysis=0.5, SQL components pre-generation=1.5), `AST_Matching_reward = 1`, `Execution_Reward = 3` (Syntax Reward = 1, Accuracy Reward = 2, Format_Reward = 1). Additional implementation details can be found in the appendix A.3.

**Baselines:** We choose three types of baseline methods for a thorough comparison. *(1) Prompting-based methods* (e.g., CHASE-SQL (Pourreza et al., 2024), MAC-SQL (Wang et al., 2024), DAIL-SQL (Gao et al., 2023)) — these comparisons show our trained model's advantage over closed models like GPT-4. *(2) SFT-based methods* (Codes (Li et al., 2024b), ROUTE (Qin et al., 2024), Omini-SQL (Li et al., 2025b), SQL-O1 (Lyu et al., 2025)) — these highlight our model's superiority over models trained by supervised fine-tuning (SFT). *(3) RL-based methods* (SENSE (Yang et al., 2024), DPO (Rafailov et al., 2023), Reasoning-SQL (Pourreza et al., 2025), Reward-SQL (Zhang et al., 2025), sql-r1 (Ma et al., 2025), Think2SQL (Papicchio et al., 2025), ReEx-SQL (Dai et al., 2025)) — these demonstrate the benefits of our GRPO rule-based training compared to other RL algorithms. Some RL-based Text-to-SQL methods like (Yao et al., 2025) use techniques such as data augmentation, making direct comparison difficult. Therefore, we only compare with methods closely related to ours.

## 3.2 RESULTS AND ANALYSIS

Table 1: Main Results(%). Results re-evaluated with the open-source code are marked with †. For open-source LLMs, the best and second-best results are shown in **bold** and underline, respectively.

| Methods | Base Model | BIRD (DEV) | | SPIDER (DEV) | SPIDER (TEST) | SPIDER (SYN) | SPIDER (REAL) |
|---|---|---|---|---|---|---|---|
| | | EX | VES | EX | EX | EX | EX |
| *Prompting-based* | | | | | | | |
| MAC-SQL(Wang et al., 2024) | GPT-4 | 59.4 | 66.2 | 86.7 | 82.8 | - | - |
| DAIL-SQL(Gao et al., 2023) | GPT-4 | 54.8 | 56.1 | 83.5 | 86.2 | - | - |
| CHASE-SQL(Pourreza et al., 2024) | Gemini-1.5 | 73.0 | 73.0 | - | 87.6 | - | - |
| *SFT-based* | | | | | | | |
| ROUTE(Qin et al., 2024) | Qwen2.5-Coder-14B | 60.8 | 65.2 | 80.9 | 87.1 | - | - |
| Codes-15B(Li et al., 2024b) | StarCoder-15B | 58.5 | 56.7 | 79.4 | - | 69.4 | 75.6 |
| Omini-SQL-14B(Li et al., 2025b) | Qwen2.5-Coder-14B | 64.2 | - | 81.4 | **88.3** | 69.0 | 76.4 |
| SQL-O1(Lyu et al., 2025) | Llama3-8B | 63.4 | 64.7 | 79.6 | 85.4 | 77.6 | **82.7** |
| *RL-Based(DPO)* | | | | | | | |
| SENSE-13B(Yang et al., 2024) | CodeLLaMA-13B | 55.5 | - | **84.1** | 86.6 | 70.2 | 76.6 |
| DPO(Rafailov et al., 2023) | Qwen2.5-Coder-7B | 64.1 | - | 82.6 | 80.2 | 76.2 | 79.1 |
| *RL-Based(GRPO)* | | | | | | | |
| SQL-R1-7B(Ma et al., 2025) | Qwen2.5-Coder-7B | 60.8† | 61.08† | 83.8† | 87.6† | 76.7† | 76.8† |
| Think2SQL-7B(Papicchio et al., 2025) | Qwen2.5-Coder-7B | 56.1 | - | - | 82.4 | 78.0 | - |
| Reasoning-SQL-7B(Pourreza et al., 2025) | Qwen2.5-Coder-7B | 64.0 | - | - | 78.7 | 78.7 | 73.3 |
| Reward-SQL-7B(Zhang et al., 2025) | Qwen2.5-Coder-7B | **66.4** | - | 77.0 | - | - | - |
| Rex-SQL-7B(Dai et al., 2025) | Qwen2.5-Coder-7B | 64.9 | **73.1** | 83.7 | 86.6 | 72.1 | 79.1 |
| **Ours** | Qwen2.5-Coder-7B | 65.8 | 66.5 | 83.0 | 84.4 | **80.7** | 79.9 |

**Main Results.** Table 1 presents our main experimental results, and we also report experimental results on Spider 2.0 to validate the effectiveness of our methods (see A.4.2 for more details). We train solely on the BIRD dataset and do not use any SPIDER data. As SQL-R1 reports only self-consistency, we reproduce their greedy search results for fair comparison. From the results, we have the following findings: (1) For open-source models, our method achieves strong overall performance across all datasets. In RL-based and SFT-based methods, our performance on BIRD is second only to Reward-SQL-7B. However, Reward-SQL-7B uses CTE data synthesis and cold start strategies, while our method trains GRPO only based on designed reward rules. Also, on the SPIDER-DEV dataset, our method (83%) significantly outperforms Reward-SQL (77%). This shows our method generalizes better than Reward-SQL. (2) On some datasets, our 7B model matches or exceeds GPT-4. On BIRD, our method surpasses MAC-SQL and DAIL-SQL. On SPIDER-DEV, it is slightly below DAIL-SQL+GPT4. (3) Our method has good generalization. We train only on the

BIRD training set, yet achieve 83% and 84.4% execution accuracy on SPIDER-DEV and SPIDER-TEST. This shows the model trained with our reasoning method can generalize to other datasets. (4) Our method is robust. On SPIDER-SYN and SPIDER-REAL, our method achieves about 80% accuracy, exceeding other models of similar size. This indicates strong robustness of our trained model. (5) SQL generated by our method has good execution efficiency. Our method's SQL execution efficiency reaches 66.49, higher than MAC-SQL and DAIL-SQL. Although lower than Rex-SQL, that method uses tree decoding and multiple interactions (10 times), resulting in higher computational cost.

Table 2: Ablation results of the reward function, where ✓ means that the item is adopted

| Reward Function | | | | | BIRD-DEV(EX%) | | | |
|---|---|---|---|---|---|---|---|---|
| base | pre-gen | ast | idf | n-gram | simple | moderate | challenging | total |
| ✓ | | | | | 69.19 | 55.91 | 55.56 | 63.89 |
| ✓ | | ✓ | | | 69.84 | 55.48 | 54.17 | 64.02 |
| ✓ | | ✓ | ✓ | | 69.73 | 55.91 | 55.56 | 64.21 |
| ✓ | ✓ | | | | 69.51 | 55.91 | 59.03 | 64.41 |
| ✓ | ✓ | | | ✓ | 68.97 | 55.70 | 57.64 | 63.80 |
| ✓ | ✓ | ✓ | | | 70.05 | 56.34 | 60.42 | 64.99 |
| ✓ | ✓ | ✓ | ✓ | | **70.38** | **57.42** | **63.19** | **65.78** |

**Ablation Study** The table 2 shows ablation results for the reward function. Here, *base* includes execution and format rewards; *pre-gen* is the SQL components pre-generation reward; *ast* denotes the AST matching reward, where idf means AST node initial values use idf scores. We also compare n-gram matching ($n = 2$, as in Reasoning-SQL) instead of AST matching. From the table, we observe: **(1)** On *base*, adding only AST matching reward (second row) increases the overall metric from 63.89 to 64.02. Adding IDF weighting on nodes (third row) further improves it to 64.21. This indicates that both AST matching reward and IDF node weighting help SQL accuracy, regardless of component pre-generation. **(2)** Adding SQL component pre-generation to *base* (fourth row) notably boosts the metric on the challenging data ($55.56\% \rightarrow 59.03\%$), showing that SQL component pre-generation is effective. **(3)** Comparing SQL component pre-generation plus n-gram matching (fifth row, 63.80) with AST matching reward (sixth row, 64.99), the latter performs better. **(4)** The last row shows that *pre-gen+ast+idf* achieves the highest result. The challenging metric increases from 60.42 (*pre-gen+ast*) to 63.19. This is because high-level SQL operations (like set operations) are rare and have higher IDF values; rewarding correct generation of these operations improves accuracy on difficult samples.

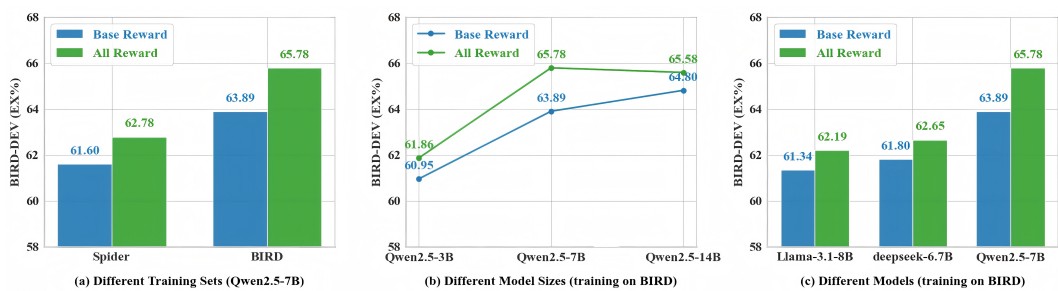

Figure 4: Generalization Experiments.

**Generalization Experiments.** To demonstrate our method's generalizability, we present results under different reward settings across three scenarios. Figure 4(a) compares Qwen2.5-Coder-7B-Instruct trained on various datasets using *base reward* (execution and format rewards) versus *all reward*. Figure 4(b) shows results for models from Qwen2.5-Coder-3B-Instruct to Qwen2.5-Coder-14B-Instruct. Figure 4(c) compares architectures including Llama3.1-8B-Instruct, deepseek-coder-6.7b-Instruct, and Qwen2.5-Coder-7B-Instruct. All use BIRD-DEV for validation. Key observations: **(1)** Adding SQL components pre-generation and AST Matching Rewards consistently improves execution accuracy on both SPIDER and BIRD datasets, showing strong generalization. **(2)**

Our method benefits all model sizes and architectures. Figures 4(b) and 4(c) show improvements over the base reward across models, with smaller gains on Qwen2.5-Coder-14B-Instruct, suggesting it is more effective for smaller models that gain more from fine-grained reasoning optimization.

**Case Study.** Figure 5 shows a case study comparing the standard CoT reasoning paradigm (middle) with the SQL components pre-generation paradigm (right). Both correctly identify at step three that the user's question needs a threshold filter. However, CoT does not generate this SQL component, so the model cannot effectively use this information when producing the final SQL, leading to errors. In contrast, the SQL component pre-generation paradigm generates the corresponding component at this step, allowing the model to use this information effectively when generating the final SQL, thus producing the correct SQL statement. We also report a case from Spider 2.0 ( A.4.3) and a more complex case involving multi-level nested SQL queries ( A.4.4) to demonstrate the effectiveness of SQL component pre-generation.

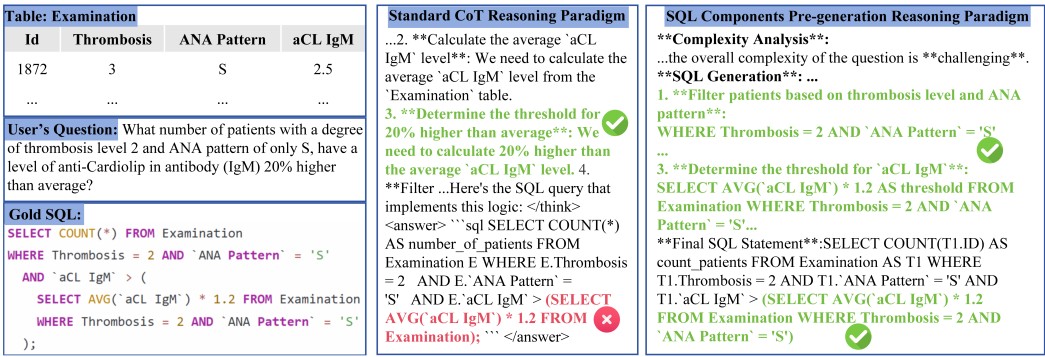

Figure 5: Case Study. Green text shows correct parts, red text shows errors.

**Impact of the Complexity Analysis Module.** We conducted four comparison experiments to study the impact of the complexity analysis module, as shown in table 3: ***w/o:*** Without the complexity analysis module. SQL component pre-generation is applied to all questions, ignoring their complexity; ***rlue-based:*** The complexity is pre-evaluated by the script and directly input as prior information during Text-to-SQL training. The model does not need to predict it; ***model-based:*** The complexity analysis module without manual intervention. The model is prompted to judge the complexity of the question by itself. ***(model&rule)-based:*** Use a complexity evaluation script to label the data in advance. During Text-to-SQL training, the model is trained to predict this complexity label.

Table 3: Impact of the Complexity Analysis Module

| Complexity Analysis Module | BIRD-DEV(EX%) | | | |
|---|---|---|---|---|
| | simple | moderate | challenging | total |
| w/o | **70.70** | 55.27 | 58.33 | 64.86 |
| rule-based | 70.49 | 56.56 | 59.72 | 65.25 |
| model-based | 70.27 | 55.91 | 57.64 | 64.73 |
| (model&rule)-based | 70.38 | **57.42** | **63.19** | **65.78** |

Table 4: Evaluation of complexity analysis

| Complexity Label | BIRD-DEV(EX%) | | |
|---|---|---|---|
| | Precision | Recall | F1 |
| simple | 0.78 | 0.75 | 0.77 |
| moderate | 0.75 | 0.75 | 0.75 |
| challenging | 0.86 | 0.88 | 0.87 |

From the results, we observe: **(1)** The worst performance occurs without (64.86) or with model-based (64.73) methods, indicating that pre-generating SQL components for all questions is ineffective. Without clear rules, the model often misjudges complexity. **(2)** Comparing rule-based and (rule&models)-based approaches, the latter performs better. This is because rule-based methods struggle with special cases where logical complexity and SQL component count do not align (e.g., a logically simple query with many components or vice versa). Thus, combining model-driven complexity judgment with rule-based methods yields more accurate complexity analysis and better SQL component pre-generation.

Table 4 shows the complexity prediction metrics. The model's F1 scores for the three difficulty labels are modest, indicating it does not rely solely on SQL component count. Since component count and

logical complexity don't always correlate, the model learns to evaluate both aspects, enabling more comprehensive complexity prediction.

**Impact of AST Matching on SQL Component Accuracy.** We further analyze the impact of AST on the EM metric to reflect improvements in local component accuracy. We compare three cases: without local component matching (first row), using n-gram for local component matching (second row), and using the AST matching method proposed in this paper (third row). To ensure a fair comparison of EM changes, we select data with similar EX values for the EM comparison. The results are shown in Table 5. It can be observed that the n-gram based local component matching significantly outperforms the case without any local component matching reward, indicating that improvements in local component accuracy are reflected in the EM metric. Furthermore, the EM scores for the AST matching method exceed those of the n-gram matching method, demonstrating the effectiveness of the proposed AST matching approach in enhancing local component accuracy.

Table 5: Comparison of Matching Methods

| Matching Methods | SPIDER-DEV | | SPIDER-TEST | |
|---|---|---|---|---|
| | EX% | EM% | EX% | EM% |
| w/o | 81.7 | 11.7 | 82.8 | 26.2 |
| n-gram | 81.7 | 37.2 | 83.3 | 36.3 |
| ast | 81.8 | 45.5 | 83.3 | 45.1 |

Table 6: AST rewards on different datasets

| Datasets | Average Reward | |
|---|---|---|
| | positive | negative |
| BIRD-DEV | 0.78 | 0.64 |
| SPIDER-DEV | 0.51 | 0.39 |
| SPIDER-TEST | 0.50 | 0.38 |

**Rationality Analysis of the AST Matching Rule.** We assess the AST matching reward's ability to distinguish positive from negative samples. Table 6 shows that positive samples have significantly higher average AST matching rewards than negative ones across three datasets. The relatively low reward on Spider likely stems from training on Bird and Spider's simpler SQL with common keywords (SELECT, WHERE, etc.), leading to lower IDF values for AST nodes and thus lower rewards. This confirms the AST matching rule's validity and reveals no reward hacking.

Table 7: Confusion matrix for complexity label prediction

| | BIRD-DEV | | |
|---|---|---|---|
| | simple | moderate | challenging |
| **simple** | 603 | 78 | 7 |
| **moderate** | 69 | 388 | 63 |
| **challenging** | 27 | 53 | 245 |

Table 8: SQL error rates for each predicted complexity category

| | BIRD-DEV | | |
|---|---|---|---|
| | simple | moderate | challenging |
| **simple** | 22.22% | 44.87% | 57.14% |
| **moderate** | 52.14% | 37.62% | 34.92% |
| **challenging** | 62.26% | 48.15% | 47.43% |

**Error Analysis of Complexity Analysis Module.** We further analyzed errors in the complexity analysis module. Table 7 shows the confusion matrix (rows: true labels; columns: predicted categories), while Table 8 reports SQL error rates by predicted category with the same row/column format. Key findings are as follows: *(1) SQL Component pre-generation is effective for moderate and challenging questions.* For example, as shown in Table 8, when the true complexity label is moderate, if it is predicted as simple, the error rate is 52.14%, whereas if it is predicted as challenging, the error rate decreases to 34.92%. Similarly, when the true label is challenging, the error rate is 62.26% if misclassified as simple, but drops to 48.15% if classified as moderate. This further demonstrates the effectiveness of component pre-generation for moderate and challenging SQL queries. This is because when moderate or challenging questions are misclassified as simple, SQL component pre-generation is not performed, leading to poorer performance. However, when moderate questions are misclassified as challenging, or challenging questions as moderate, SQL component pre-generation is still applied, resulting in lower error rates. *(2) Not all queries require component pre-generation.* For example, if a simple query is misclassified as moderate or challenging, unnecessary pre-generation steps are performed, resulting in higher SQL error rates (44.87% and 57.14% respectively). This is consistent with the conclusions in Table 3 of the paper. *(3) Misclassification often occurs between adjacent complexity levels.* For example, for simple queries, most misclassifications are as moderate (78 cases), while the misclassification as challeng-

ing is much less frequent (7 cases). Similarly, for challenging queries, most misclassifications are as moderate (53 cases), and only a few are as simple (27 cases). This shows that the main difficulty in complexity classification lies between adjacent categories. Therefore, an alternative approach based on a continuous complexity score or a hierarchical classification method might be worth investigating in future work.

**Efficiency Evaluation.** While our method introduces complexity analysis and SQL components pre-generation steps, we still adopt single-round training and inference. Therefore, the overhead is comparable to the standard GRPO method. We provide comparisons of various consumption metrics. As shown in Table 9, compared to the standard GRPO training method (execution accuracy reward and format reward), our method only increases token consumption by 0.23k but improves execution accuracy (EX) by 1.89. Compared to MCTS-based methods such as Alpha-SQL, our computational cost and inference latency are significantly lower.

Table 9: Comparison of different methods on various consumption metrics

| Methods | Models | EX | Token Usage (k tokens) | | | Latency (s/step) | |
|---------|--------|-----|------|--------|-------|----------|-----------|
| | | | Input | Output | Total | Training | Inference |
| Alpha-SQL | | 66.8 | 138 | 72.2 | 200.2 | - | 377.1 |
| GRPO | Qwen2.5-Coder-7B | 63.89 | 0.38 | 0.29 | 0.67 | 250.82 | 3.05 |
| **Ours** | | 65.78 | 0.42 | 0.48 | 0.90 | 267.73 | 5.08 |

## 4    RELATED WORK

**LLM Reasoning.** Inspired by the CoT series (Wei et al., 2022), researchers recognize the importance of enhancing reasoning in LLMs. Current methods fall into two categories: (1) Training-free methods include Prompt Strategy, which guides step-by-step reasoning via prompts (Wei et al., 2022; Zhou et al., 2022; Wang et al., 2022), and Search Strategy like MCTS (Coulom, 2006; Zhang et al., 2023; Tian et al., 2024), which performs one-step lookahead. (2) Training-based methods mainly use reinforcement learning (Rafailov et al., 2023; Shao et al., 2024; Schulman et al., 2017); for instance, DeepSeek-R1 (Guo et al., 2025) uses GRPO to achieve strong reasoning.

**LLM Reasoning for Text-to-SQL.** Text-to-SQL converts natural language queries to SQL, enabling database interaction. Improving LLM reasoning in this field involves both training-free and training-based methods: **(1) Training-free:** Works such as (Pourreza & Rafiei, 2024a; Gao et al., 2023; Wang et al., 2024; Dong et al., 2023) employ prompts to encourage longer reasoning chains for better SQL generation. Later methods like MCTS-SQL (Yuan et al., 2025) and Alpha-SQL (Li et al., 2025a) apply MCTS to boost reasoning. **(2) Training-based:** Early efforts used supervised fine-tuning (SFT) (Li et al., 2024b; 2025b; Pourreza & Rafiei, 2024b; Lyu et al., 2025). With reinforcement learning's rise, approaches shifted toward it. For example, SENSE (Yang et al., 2024) combines weak and strong models with DPO (Rafailov et al., 2023) for improved results. GRPO (Shao et al., 2024), which allows faster rule-based rewards without extra reward models, is widely adopted. Reasoning-SQL (Pourreza et al., 2025) designs six reward rules for training; SQL-R1 (Ma et al., 2025) studies reward rules and cold start effects; Arctic-Text2SQL-R1 (Yao et al., 2025) and COGNISQL-R1-ZERO (Gajjar et al., 2025) focus on data filtering and execution correctness respectively; Think2SQL (Papicchio et al., 2025) examines fine-grained rewards. Most focus on output rewards, while Reward-SQL (Zhang et al., 2025) proposes a process reward with CTE expressions. Besides SQL generation, CSC-SQL trains SQL revision models using GRPO.

## 5    CONCLUSION AND FUTURE WORK

This work tackles the semantic blind spot in pending SQL components during complex Text-to-SQL generation. We propose a complexity-guided pre-generation approach and a maximum connected subtree matching reward based on SQL AST to improve local accuracy. Experiments validate our method's effectiveness. However, AST matching shows limited generalization and misses some semantic cases. Future work will focus on enhancing its generalization to boost performance.

## 6 ETHICS STATEMENT

This study adheres to the ethical guidelines of ICLR. No human participants or animal experiments were involved in this research. All datasets used, were obtained in accordance with relevant usage regulations to ensure no violation of privacy. We took care to avoid any biased or discriminatory outcomes throughout the study. No personally identifiable information was used, nor were any experiments conducted that could raise privacy or security concerns. We are committed to maintaining transparency and integrity throughout the entire research process.

## 7 REPRODUCIBILITY STATEMENT

We have made every effort to ensure that the results presented in this paper are reproducible. The paper provides detailed descriptions of the experimental setup, including training procedures, model configurations, and hardware environments.

## 8 USE OF LARGE LANGUAGE MODELS (LLMS)

Large language models (LLMs) were utilized in the drafting and polishing stages of this paper. Specifically, LLMs assisted in improving language expression, enhancing readability, and ensuring clarity across various sections of the manuscript. Tasks such as sentence restructuring, grammar checking, and overall text fluency enhancement were supported by the model.

It is important to emphasize that LLMs did not participate in the conceptualization, research design, or experimental procedures of this study. All research ideas, designs, and analyses were independently developed by the authors. The contribution of LLMs is limited solely to improving the language quality of the paper and does not involve scientific content or data analysis.

The authors take full responsibility for the content of the paper, including the text generated or refined with the aid of LLMs. We ensure that all LLM-generated content complies with ethical standards and does not involve plagiarism or scientific misconduct.

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

# A APPENDIX

## A.1 DATASETS:

This paper mainly utilizes the Spider(Yu et al., 2018) and Bird(Li et al., 2024c) datasets, which are the most widely used datasets for the Text-to-SQL task. The Spider dataset contains 8,659 training examples, 1,034 development examples, and 2,147 test examples. The Bird dataset is more complex than Spider and better reflects real-world scenarios, comprising 9,428 training examples and 1,534 development examples.

In addition, this paper also employs two variants of the Spider dataset: Spider-Syn(Gan et al., 2021) and Spider-Real(Deng et al., 2020), both derived from the Spider-dev dataset.

Spider-Syn replaces natural language queries with synonyms; for example, the query "what is the type of the document named 'David CV'" becomes "what is the type of the file named 'David CV'" in Spider-Syn. This approach breaks the direct correspondence between natural language queries and database fields to simulate attacks on schema linking, thereby challenging the model's robustness. Spider-Real not only replaces or removes column names but also substitutes phrases in natural language queries; for instance, "How many pets have a greater weight than 10" is changed to "How many pets are over 10 lbs?" Therefore, we select these two datasets to evaluate the robustness of our model.

## A.2 EVALUATION METRICS:

We use execution accuracy (EX) as the evaluation metric, which considers a prediction correct if the execution results of the predicted SQL and the gold SQL match, and incorrect otherwise. We also use VES to evaluate SQL execution efficiency, where VES measures the average execution time after running the SQL multiple times. Additionally, we assess the accuracy of SQL components using exact match (EM), which is based on literal matching. For the BIRD dataset, we apply the official execution accuracy script, while for the SPIDER dataset and its variants, we use the `spider-test-suite` script.

## A.3 IMPLEMENTATION DETAILS:

Our experiments mainly use Qwen2.5-Coder-7B-Instruct (Hui et al., 2024). To test generality, we also show results on 3B and 14B models, and other models like Llama-3.1-8B-Instruct (Grattafiori et al., 2024) and deepseek-coder-6.7b-Instruct (Guo et al., 2024). We apply reinforcement learning with the VERL[1] framework. Key settings are: `max_prompt_length` = 4096, `max_response_length` = 2048, `rollout_n` = 5, `ppo_micro_batch_size_per_gpu` = 8, learning rate = $1 \times 10^{-6}$, and `total_epochs` = 30; others use default values. Reward weights are: `reasoning_steps_reward` = 2 (Complexity analysis = 0.5, Sub-SQL pre-generation = 1.5), `AST_Matching_reward` = 1, `Execution_Reward` = 3 (Syntax Reward = 1, Accuracy Reward = 2, Format_Reward = 1). Experiments ran on 8 NVIDIA A800-SXM4-80GB GPUs, with about 30 hours training for the 7B model. All syntax tree constructions use the SQLglot open-source library[2], a powerful SQL parser.

---

[1]https://github.com/volcengine/verl
[2]https://github.com/tobymao/sqlglot

## A.4 SUPPLEMENTARY EXPERIMENTS

Table 10: Reward Function Weight Search

| reward function weights | | | | | BIRD-DEV |
| --- | --- | --- | --- | --- | --- |
| syntax | acc | format | ast | reason | EX |
| 1 | 1 | 1 | 0 | 0 | 63.89 |
| 1 | 1 | 1 | 1 | 1 | 65.45 |
| 1 | 2 | 1 | 1 | 1 | 65.78 |
| 1 | 1 | 1 | 2 | 1 | 64.86 |
| 1 | 1 | 1 | 1 | 2 | 64.86 |

### A.4.1 REWARD FUNCTION WEIGHT SEARCH

We analyzed the impact of different reward function weights on the results. When the weights for AST matching and reasoning step rewards increase to 1, execution accuracy improves from 63.89 to 65.45, showing their effectiveness. When the accuracy reward weight increases to 2, execution accuracy further rises to 65.78, indicating that a reasonable weight setting gives higher priority to the execution accuracy reward. However, when the weights for AST matching and reasoning step rewards both increase to 2, execution accuracy decreases, possibly because these rewards become too dominant, weakening the impact of the execution accuracy reward and affecting overall performance. Still, the accuracy remains significantly higher than 63.89%, showing that our method does not fully rely on the specific design of the reward function weights.

### A.4.2 EVALUATION ON SPIDER2.0

We have now supplemented our experiments on Spider2.0. Specifically, we selected Qwen2.5-Coder-7B-Instruct and OminiSQL-7B as base models. The experimental results are shown in the table, where "base rewards" refers to the standard GRPO reward design, which includes only execution result rewards and format rewards. As shown, the Qwen model achieves an execution accuracy of only 1.5% on Spider2.0. After applying "base rewards," the accuracy rises to 5.9%, demonstrating that the GRPO algorithm significantly improves the model's ability to generate SQL on this challenging benchmark. With our proposed SQL components pre-generation and maximum connected subtree matching reward mechanisms, the execution accuracy further increases by 2.9% (from 5.9% to 8.8%), verifying that our method remains effective on Spider2.0. Similarly, on OminiSQL-7B, our method improves accuracy by 3% over the standard GRPO algorithm, and the gap with GPT-4o and DeepSeek-V3 is reduced to only 1.5%.

This demonstrates the ability of our method to generate complex SQL queries and its effectiveness on more challenging datasets such as Spider 2.0.

Table 11: Peformance on Spider 2.0

| Model | Reward Function | Spider-2.0-SQLite (EX%) |
| --- | --- | --- |
| GPT-4o | - | 15.6 |
| DeepSeek-V3 | - | 15.6 |
| | | |
| Qwen2.5-Coder-7B-Instruct | - | 1.5 |
| Qwen2.5–Coder-7B-Instruct+GRPO | Base Rewards | 5.9 |
| **Qwen2.5-Coder-7B-Instruct+GRPO (Ours)** | All Rewards | **8.9** |
| | | |
| Omini-SQL-7B | - | 10.4 |
| Omini-SQL-7B+GRPO | Base Rewards | 11.1 |
| **Omini-SQL-7B+GRPO (ours)** | All Rewards | **14.1** |

### A.4.3   CASE STUDY ON SPIDER 2.0

To further evaluate the effectiveness of our method, we compare the SQL component pre-generation reasoning paradigm with the Chain-of-Thought (CoT) reasoning paradigm using a highly challenging real-world case from Spider 2.0. This case involves parallel and multi-level nested SQL structures.

*User's Question: Could you find out the average percentage of the total lifetime sales (LTV) that occur in the first 7 and 30 days after a customer's initial purchase? Also, include the average total lifetime sales (LTV). Please exclude customers with zero lifetime sales. The 7- and 30-day periods should be based on the exact number of hours-minutes-seconds, not calendar days.*

First, we examine the result obtained by the CoT reasoning paradigm:

**(1) Chain-of-Thought (CoT) Reasoning Paradigm:**

To answer this question, we need to calculate the average percentage of the total lifetime sales (LTV) that occur in the first 7 and 30 days after a customer's initial purchase. We also need to include the average total lifetime sales (LTV) and exclude customers with zero lifetime sales.

Here is the step-by-step reasoning process:

1. Identify the relevant tables: `payment` and `customer`.

2. Join the tables via the `customer_id`.

3. Calculate total lifetime sales (LTV) for each customer.

4. Filter payments within the first 7 and 30 days of the customer's initial purchase.

5. Compute the average percentage of LTV in these periods.

6. Exclude customers with zero LTV.

Here is the SQL query that corresponds to this reasoning process:

```sql
SELECT
  CAST(SUM(CASE WHEN julianday(T1.payment_date) - julianday(T2.
      create_date) BETWEEN 0 AND 7 THEN T1.amount ELSE 0 END) AS REAL) *
      100 / SUM(T1.amount) AS avg_7_day_sales,
  CAST(SUM(CASE WHEN julianday(T1.payment_date) - julianday(T2.
      create_date) BETWEEN 0 AND 30 THEN T1.amount ELSE 0 END) AS REAL) *
       100 / SUM(T1.amount) AS avg_30_day_sales,
  SUM(T1.amount) AS avg_ltv
FROM
  payment AS T1
JOIN
  customer AS T2 ON T1.customer_id = T2.customer_id
WHERE
  T1.amount > 0;
```

> **Case Analysis**
>
> It can be seen that the CoT reasoning paradigm reasonably decomposes the complex problem into necessary sub-steps. However, when generating the final SQL, some issues arise in translating these subtasks. For example, in step 3, the model correctly decomposes the task to calculate the total lifetime sales (LTV) but fails to group data by customer in the final SQL.

Next, we examine the result generated by the SQL component pre-generation reasoning paradigm:

**(2) SQL Component Pre-Generation Reasoning Paradigm:**

**Complexity Analysis:** The question requires multiple aggregations and conditional logic: calculating total LTV for each customer, finding the first purchase date, summing sales in the first 7 and 30 days, and finally computing the average percentage and average LTV, while excluding customers

with zero LTV. This involves multi-level nesting and subqueries. Thus, the overall complexity is **challenging**.

**Step-by-step SQL decomposition:** To calculate the average percentage of the total lifetime sales (LTV) that occur in the first 7 and 30 days after a customer's initial purchase, we need to follow these steps:

1. **Step 1:** Calculate the total lifetime sales (LTV) for each customer.

```sql
SELECT
    c.customer_id,
    SUM(p.amount) AS total_ltv
FROM customer c
JOIN payment p ON c.customer_id = p.customer_id
GROUP BY c.customer_id
```

2. **Step 2:** Determine the first purchase date for each customer.

```sql
SELECT
    c.customer_id,
    MIN(p.payment_date) AS first_purchase_date
FROM customer c
JOIN payment p ON c.customer_id = p.customer_id
GROUP BY c.customer_id
```

3. **Step 3:** Calculate the sales in the first 7 and 30 days after the first purchase.

```sql
SELECT
    c.customer_id,
    SUM(CASE WHEN julianday(p.payment_date) - julianday(fp.
        first_purchase_date) BETWEEN 0 AND 7 THEN p.amount ELSE 0
        END) AS ltv_7_days,
    SUM(CASE WHEN julianday(p.payment_date) - julianday(fp.
        first_purchase_date) BETWEEN 0 AND 30 THEN p.amount ELSE 0
        END) AS ltv_30_days,
    SUM(p.amount) AS total_ltv
FROM customer c
JOIN payment p ON c.customer_id = p.customer_id
JOIN (
    SELECT
        c.customer_id,
        MIN(p.payment_date) AS first_purchase_date
    FROM customer c
    JOIN payment p ON c.customer_id = p.customer_id
    GROUP BY c.customer_id
) fp ON c.customer_id = fp.customer_id
GROUP BY c.customer_id
```

**Final SQL statement:**

```sql
SELECT
    AVG(ltv_7_days / total_ltv * 100) AS avg_ltv_7_days,
    AVG(ltv_30_days / total_ltv * 100) AS avg_ltv_30_days,
    AVG(total_ltv) AS avg_total_ltv
FROM (
    SELECT
        c.customer_id,
        SUM(CASE WHEN julianday(p.payment_date) - julianday(fp.
            first_purchase_date) BETWEEN 0 AND 7 THEN p.amount ELSE 0 END
            ) AS ltv_7_days,
        SUM(CASE WHEN julianday(p.payment_date) - julianday(fp.
            first_purchase_date) BETWEEN 0 AND 30 THEN p.amount ELSE 0
            END) AS ltv_30_days,
        SUM(p.amount) AS total_ltv
```

```
    FROM customer c
    JOIN payment p ON c.customer_id = p.customer_id
    JOIN (
        SELECT
            c.customer_id,
            MIN(p.payment_date) AS first_purchase_date
        FROM customer c
        JOIN payment p ON c.customer_id = p.customer_id
        GROUP BY c.customer_id
    ) fp ON c.customer_id = fp.customer_id
    GROUP BY c.customer_id
) subquery
WHERE total_ltv > 0;
```

**Case Analysis**

This is a complex SQL statement with two levels of nesting, involving multiple logical operations and nested structures. Compared to the CoT reasoning paradigm, the SQL component pre-generation paradigm not only decomposes the task step-by-step but also simultaneously generates corresponding SQL statements, thereby reducing the complexity of the task. For example, in the first step, the model identifies the need to "Calculate the total lifetime sales (LTV) for each customer," which corresponds to step 3 in the CoT reasoning paradigm, and generates the related SQL fragment accordingly. This greatly reduces the possibility of errors in generating the final SQL query for that subtask.

### A.4.4 CASE STUDY ON MULTI-LEVEL NESTED SQL

Our method has a certain capability for pre-generating components for multi-level nested structures. We take the example of an extremely complex SQL query with parallel and three-level nesting. We still compare the SQL component pre-generation reasoning paradigm and the CoT reasoning paradigm.

*User's Question: Find the names of schools whose enrollment (K-12) is higher than the average enrollment (K-12) of schools in the district that has the highest average SAT Math score.*

Let's first look at the generation results of the SQL component pre-generation reasoning paradigm.

**(1) SQL Component Pre-Generation Reasoning Paradigm:**

**Complexity Analysis:** The question requires multiple steps: calculating the average SAT Math score by district, finding the district with the highest average SAT Math score, calculating the average enrollment for schools in that district, and then finding the names of schools whose enrollment is higher than this average. This involves multiple joins, aggregations, and subqueries. Therefore, the overall complexity of the question is **challenging**.

**SQL Generation:** To find the names of schools whose enrollment (K-12) is higher than the average enrollment (K-12) of schools in the district that has the highest average SAT Math score, we need to follow these steps:

**Step-by-step SQL decomposition:**

1. **Step 1:** Calculate the average SAT Math score by district.

```
SELECT s.District, AVG(ss.AvgScrMath) AS avg_math_score
FROM schools s
JOIN satscores ss ON s.CDSCode = ss.cds
GROUP BY s.District
```

2. **Step 2:** Find the district with the highest average SAT Math score.

```
SELECT s.District
FROM schools s
```

```
JOIN satscores ss ON s.CDSCode = ss.cds
GROUP BY s.District
ORDER BY AVG(ss.AvgScrMath) DESC
LIMIT 1
```

3. **Step 3:** Calculate the average enrollment for schools in the district with the highest average SAT Math score.

```
SELECT AVG(f."Enrollment␣(K-12)") AS avg_enrollment
FROM frpm f
JOIN schools s ON f.CDSCode = s.CDSCode
WHERE s.District = (
    SELECT s.District
    FROM schools s
    JOIN satscores ss ON s.CDSCode = ss.cds
    GROUP BY s.District
    ORDER BY AVG(ss.AvgScrMath) DESC
    LIMIT 1
)
```

**Final SQL statement:**

```
SELECT s.`School Name`
FROM frpm f
JOIN schools s ON f.CDSCode = s.CDSCode
WHERE s.District = (
    SELECT s.District
    FROM schools s
    JOIN satscores ss ON s.CDSCode = ss.cds
    GROUP BY s.District
    ORDER BY AVG(ss.AvgScrMath) DESC
    LIMIT 1
)
AND f."Enrollment␣(K-12)" > (
    SELECT AVG(f."Enrollment␣(K-12)")
    FROM frpm f
    JOIN schools s ON f.CDSCode = s.CDSCode
    WHERE s.District = (
        SELECT s.District
        FROM schools s
        JOIN satscores ss ON s.CDSCode = ss.cds
        GROUP BY s.District
        ORDER BY AVG(ss.AvgScrMath) DESC
        LIMIT 1
    )
)
```

**Case Analysis**

From the generation process above, it is clear that step 2 serves as a pre-generated sub-SQL for step 3, and step 3 is a pre-generated sub-SQL for the final SQL. This indicates that our method has a certain supporting effect for multi-level nesting. Additionally, step 1 is also a sub-SQL of the final SQL, and it is parallel in structure with step 3. This shows that our method effectively pre-generates sub-SQL in parallel structures as well.

Let's also look at the results obtained from the CoT reasoning paradigm to further illustrate the effectiveness of SQL component pre-generation:

**(2) CoT Reasoning Paradigm**

**Here's a step-by-step breakdown:**

1. **Identify the Tables and Columns:** We need data from the frpm table for enrollment and the satscores table for SAT Math scores.

2. **Join the Tables:** We need to join the frpm table with the satscores table on the CD-SCode column to combine enrollment data with SAT scores.

3. **Calculate the Average Enrollment:** We need to calculate the average enrollment (K-12) for each district based on the satscores table.

4. **Order and Limit:** We need to order the results by the average SAT Math score in descending order to find the district with the highest average score and then limit the results to that district.

5. **Filter by Enrollment:** Finally, we filter the schools in that district to find those with enrollment (K-12) higher than the average enrollment of that district.

**Here's the SQL query that corresponds to these steps:**

```sql
SELECT T1.School
FROM frpm AS T1
INNER JOIN satscores AS T2 ON T1.CDSCode = T2.cds
WHERE T1.`Enrollment (K-12)` > (
    SELECT AVG(T1.`Enrollment (K-12)`)
    FROM frpm AS T1
    INNER JOIN satscores AS T2 ON T1.CDSCode = T2.cds
    WHERE T1.`Enrollment (K-12)` > (
        SELECT AVG(Enroll12)
        FROM satscores
    )
)
```

---

**Case Analysis**

Taking the third step as an example, the model successfully decomposed the task of calculating the average enrollment (K-12) for each district based on the satscores table. However, there was an error in the SQL generated for this step in the final SQL (the subtask mentioned calculating the average enrollment (K-12) for each district, but the generated final SQL did not reflect this). Through these two cases, on the one hand, we demonstrate that our method has a certain effectiveness for multi-level nested structures; on the other hand, it also highlights that performing SQL pre-generation during task decomposition can enhance the accuracy of the final SQL to some extent, thus affirming the effectiveness of SQL component pre-generation.

---

### A.4.5 COMPARISON OF SQL COMPONENT PRE-GENERATION AND COT REASONING PARADIGMS

**1. Explanation of Differences from CoT**

In our paper, Figures 1 and 5 illustrate the difference between the standard Chain-of-Thought (CoT) paradigm and the SQL Components Pre-generation reasoning paradigm. Below, we use Figure 5 as an example to further detail the differences and advantages of our approach, aiming to clarify our methodology.

To better explain, we first define two concepts: Task Units and SQL Units.

- ***Task Unit:*** A subtask obtained by decomposing the original question.
- ***SQL Unit:*** The SQL component that solves the corresponding task unit.

Consider the user query: *"What number of patients with a degree of thrombosis level 2 and ANA pattern of only S, have a level of anti-Cardiolipin antibody (IgM) 20% higher than average?"*

***Chain-of-Thought Reasoning Paradigm:***

- Task Unit 1: Calculate the average aCL IgM level. We need to calculate the average aCL IgM level from the Examination table.

- Task Unit 2: Determine the threshold for 20% higher than average. We need to calculate 20% higher than the average aCL IgM level.

- ...

- Final SQL:

```sql
SELECT COUNT(*) AS number_of_patients
FROM Examination E
WHERE E.Thrombosis = 2
  AND E.ANA_Pattern = 'S'
  AND E.aCL_IgM > (SELECT AVG(aCL_IgM) * 1.2 FROM Examination);
```

*SQL Components Pre-generation Reasoning Paradigm:*

- Task Unit 1: Filter patients based on thrombosis level and ANA pattern.

- SQL Unit 1: `WHERE Thrombosis = 2 AND ANA_Pattern = 'S'`

- Task Unit 2: Determine the threshold for aCL IgM.

- SQL Unit 2: `SELECT AVG(aCL_IgM) * 1.2 AS threshold FROM Examination WHERE Thrombosis = 2 AND ANA_Pattern = 'S'`

- ...

- Final SQL:

```sql
SELECT COUNT(T1.ID) AS count_patients
FROM Examination AS T1
WHERE T1.Thrombosis = 2 AND T1.ANA_Pattern = 'S'
  AND T1.aCL_IgM > (SELECT AVG(aCL_IgM) * 1.2
                    FROM Examination
                    WHERE Thrombosis = 2 AND ANA_Pattern = 'S')
```

---

**Difference Explanation**

It can be observed that the CoT decomposition typically produces only task units without generating the corresponding SQL units to solve those tasks. The disadvantage of this approach is that when generating the final SQL statement, the model only has access to the semantic information of the task units, while all SQL components remain invisible. Since task units are usually expressed in natural language, there exists a semantic gap between these natural language task units and the structured SQL units. This gap forces the model, when generating the final SQL, to both infer the SQL units corresponding to each task unit and consider how to compose these SQL units correctly, increasing the complexity of the task.

This challenge is particularly pronounced for complex SQL queries. Even if the task units are correctly decomposed, the semantic gap between natural language and SQL units often leads to difficulties in correctly converting all task units into accurate SQL units and combining them appropriately.

For example, in this case, the CoT paradigm correctly decomposes the task "We need to calculate 20% higher than the average aCL IgM level." However, the final SQL generated from all task units contains errors.

Our approach further optimizes the reasoning process specifically for the Text-to-SQL task. While decomposing the question into task units, it simultaneously generates the corresponding SQL units that solve each task unit. Therefore, when generating the final SQL, the model can leverage the semantic information of already generated SQL units. This strategy bridges the gap between natural language task units and structured SQL statements. Consequently, the model only needs to consider how to combine different SQL units when generating the final SQL, without having to infer each task unit's SQL counterpart at that stage. Compared with the CoT paradigm, this reduces the complexity of the Text-to-SQL task even further.

We formalize the two task paradigms as follows:

- **CoT formalization:** Given the generated sequence $(t_1, t_2, t_3, \ldots, t_n)$, the sequence to be generated is the final SQL $(s_1, s_2, \ldots, s_n)$. The final SQL generation can be represented as:

$$SQL = (s_1, s_2, \ldots, s_n) = f(t_1, t_2, \ldots, t_n)$$

- **Our method:** Given the generated sequence $(t_1, s_1, t_2, s_2, \ldots, t_n, s_n)$, the sequence to be generated is the final SQL $(s_1, s_2, \ldots, s_n)$. The final SQL generation is:

$$SQL = (s_1, s_2, \ldots, s_n) = f(t_1, s_1, t_2, s_2, \ldots, t_n, s_n)$$

where $f$ denotes LLMs, $t$ denotes sub-tasks, and $s$ denotes SQL components.

**2. Experimental Validation**

We further validated our method by comparing the two reasoning paradigms on the bird-dev dataset of DeepSeek-v3.2 through prompting. The results demonstrate that the SQL components pre-generation reasoning paradigm significantly outperforms the CoT paradigm in the Text-to-SQL task, summarized as follows:

Table 12: Performance comparison between CoT and SQL Components Pre-generation paradigms.

| Methods | Models | BIRD-DEV | | | |
|---|---|---|---|---|---|
| | | Simple | Moderate | Challenging | Total |
| DeepSeek-v3.2 | CoT | 62.16 | 42.73 | 36.11 | 53.72 |
| DeepSeek-v3.2 | SQL Components Pre-generation | **62.92** | **42.80** | **39.58** | **54.63** |

# B    LIMITATIONS AND FUTURE WORK

Although the complexity-guided SQL components pre-generation inference paradigm and the AST-based maximum connected subtree matching reward mechanism can effectively improve the execution accuracy of generating complex SQL queries, there are still limitations that need to be addressed in future work.

1. As mentioned in the *Error Analysis of the Complexity Analysis Module*, correctly classifying the complexity of a question can significantly enhance the accuracy of SQL generation. However, misclassification of complexity still occurs, leading to a decline in execution accuracy. Therefore, more precise complexity estimation methods warrant further exploration in future research.

2. As shown in Figure 5 and Table 3, the SQL components pre-generation paradigm enables the model to leverage the information of pre-generated SQL components when generating the final SQL, thereby improving the execution accuracy of complex SQL generation. Nevertheless, if incorrect SQL components are generated, they may also introduce interfering noise. Hence, in future work, it is crucial to optimize the accuracy of SQL component generation to reduce the interference caused by noisy components. In our work, we employ the maximum connected subtree matching reward mechanism to enhance the accuracy of local SQL components, and experiments demonstrate that this is an effective approach. However, rule-based matching rewards are limited in covering all semantically equivalent SQL variants. Therefore, future research may consider incorporating methods such as llm-as-judge(Li et al., 2024a), or executing SQL components directly on databases to obtain query results, and then setting reward values based on result comparisons—this is a promising direction for further study.

# C    GUIDELINE FOR REVIEWERS

In the revised manuscript, we use different font colors to highlight the modifications made in response to each reviewer's comments, as detailed below.

- Reviewer jJZB: •
- Reviewer mVHC: • and •
- Reviewer TWiP: • and •
- Reviewer H8Vy: • and •

