# OpenReview forum: "Improving Complex SQL Generation for Text-to-SQL by Addressing Semantic Blind Spots in Pending SQL Components"
_ICLR.cc/2026/Conference — ICLR 2026 Conference Withdrawn Submission_

### Official Review · Reviewer_H8Vy · 2025-10-28

**Soundness:** 3
**Presentation:** 2
**Contribution:** 3
**Rating:** 6
**Confidence:** 3

**Summary:**

In this paper, the authors identify a semantic blind spot in conventional Text-to-SQL generation methods. They argue that auto-regressive generation, often based on reasoning approaches like Chain-of-Thought (CoT), fails to account for upcoming SQL components, leading to errors in complex query generation. To address this, the authors propose a novel generation framework that first measures the complexity of the target SQL query. For low-complexity queries, the model employs a standard auto-regressive approach. However, for high-complexity queries, it adopts a method that pre-generates individual SQL components before integrating them into the final statement. This strategy is inspired by how humans compose complex SQL, such as nested queries, by first constructing parts of the query and then combining them.

To train a model to operate in this manner, the authors utilize reinforcement learning (RL) fine-tuning with the GRPO algorithm. They design a sophisticated reward system comprising four distinct functions: a Reasoning Steps Reward(with sub-rewards for complexity analysis and component pre-generation), an AST Matching Reward (which compares the generated SQL's Abstract Syntax Tree to the ground truth), an Execution Reward, and a Format Reward. Notably, the AST matching process incorporates a normalization step to handle discrepancies arising from aliases or differing orders, ensuring a more accurate reward calculation.

The authors evaluate their method on several Text-to-SQL benchmarks, including BIRD, SPIDER, SPIDER-SYN, and SPIDER-REAL. The proposed model achieves performance that is comparable to or surpasses state-of-the-art methods based on open-source LLMs, achieving SOTA results on SPIDER-SYN. A key finding is the model's strong generalization capability, as it was trained exclusively on the BIRD training set yet performed well on other benchmarks.

The paper includes a thorough set of empirical analyses. An ablation study on the reward functions demonstrates that the pre-generation reward significantly improves performance, particularly on challenging queries. The authors also provide a detailed analysis of the complexity analysis module, comparing its performance against several baselines . Furthermore, they demonstrate the method's versatility by applying it to diverse Large Language Models (LLMs), showing consistent performance improvements across all evaluated models. Their analysis also covers the impact of different matching reward functions (no matching, n-gram, and AST matching), confirming the superiority of the proposed AST-based approach. Finally, an ablation study on the reward function weights is detailed in the appendix.

**Strengths:**

- **Novel Problem Formulation and an Intuitive Solution:** The paper clearly articulates a significant limitation of standard auto-regressive models in Text-to-SQL generation, which it terms the "semantic blind spot". This is a well-grounded and reasonable critique based on practical SQL implementation. The proposed solution, inspired by human cognition, involves hierarchically generating SQL components before assembling them. This approach presents a more plausible and rational generation process for complex queries compared to purely auto-regressive methods.

- **Strong Generalization Performance and Thorough Experimental Analysis:** The authors convincingly demonstrate the strong generalization capabilities of their method. The paper is supported by a comprehensive set of experiments, including detailed ablation studies on the reward functions, the complexity analysis module, the matching method, and the reward weights. The analysis is further strengthened by showing consistent performance gains across various base LLM architectures.

- **Adherence to Conference Guidelines:** The authors have provided clear and commendable statements on Ethics, Reproducibility, and the Use of LLMs, following the conference's recommendations.

**Weaknesses:**

- **Clarity and Presentation:** The manuscript could benefit from a thorough round of proofreading and revision to improve its overall quality and clarity. I identified several areas for improvement:

    - **Typographical and Formatting Errors:** The text contains several typographical errors (e.g., ”semantic…” instead of “semantic…” in line 52) and inconsistent formatting, such as missing spaces in citations (e.g., 'SPIDER-SYN(...)' in line 256) and appendix references (e.g., 'appendixD.2' in line 262).

    - **Undefined Terminology and Missing Citations:** Several key terms are introduced without proper explanation or citation upon their first appearance. For instance:
	    - The SPIDER benchmark (line 92) and the GRPO algorithm (line 99) are mentioned without initial references or brief descriptions.
	    - The "Spider dataset’s method" for complexity classification (line 174) should be briefly explained.
	    - Technical terms such as IDF (line 197) and VES (line 260) should be clearly defined for the reader.
	    - The rationale for evaluating on SPIDER-SYN and SPIDER-REAL (line 256) is not provided, leaving their specific purpose and relevance unclear.

- **A Dichotomous Approach to a Continuous Problem:** The proposed method relies on a binary classification of queries into "simple" and "complex," applying distinct generation processes to each. However, query complexity is inherently a spectrum rather than a dichotomy. This rigid, two-pronged approach may lack the scalability to handle extremely complex queries that might require a more deeply nested generation process (e.g., "pre-generation of pre-generated components"). A discussion of this limitation, perhaps in a dedicated "Limitations" section, would strengthen the paper.

**Questions:**

- For simple queries where pre-generation is not performed, is the pre-generation reward simply set to zero?

- The pre-generation reward increases with the number of components generated. What was the motivation for this design choice? How does this reward function avoid incentivizing the model to generate unnecessarily verbose or redundant components?

- Could the authors elaborate on the specific characteristics of the SPIDER-SYN and SPIDER-REAL benchmarks compared to the standard SPIDER dataset? Why were these specific benchmarks chosen to validate the model's robustness?

- As a minor formatting suggestion, should Appendices A and B be moved to appear after the main references, in accordance with the author guidelines?

---

> ### Author Response · Authors · 2025-11-21
>
> Thank you for your detailed and insightful review of our paper. Below, we address each point in detail:
> ## Response to Weakness 1
> Thank you very much for your valuable feedback. We will thoroughly review and verify the formatting and wording throughout the manuscript, and update the paper accordingly before the discussion deadline.
>
> ## Response to Weakness 2
> You are absolutely right. For extremely complex SQL queries containing multiple components, using overly fine-grained component pre-generation may lead to excessively long reasoning chains, while using overly coarse-grained pre-generation may lose the semantic information of fine-grained components. Therefore, choosing an appropriate granularity for pre-generation is crucial. To address this issue, we will add a "Limitations" section in the paper to discuss in detail the limitations of the component pre-generation approach when dealing with extremely complex SQL queries, as well as potential future research directions. We expect to complete the paper update by November 25 at the latest, and certainly before the end of the discussion period.
>
> ## Response to Question 1
>
> Not exactly. when implementing the reasoning step reward, we designed two approaches. For simple queries, the step reward is consistent with the setting in DeepSeek-R1[1]: as long as the model performs reasoning by decomposing the query into subtasks, it can receive the reasoning step reward without being required to explicitly generate the corresponding SQL components for each subtask.
>
> For moderate and challenging queries, on top of the basic step reward, the model is required to generate the corresponding SQL components that solve each decomposed subtask.
>
> Moreover, to prevent reward hacking related to the choice of reasoning approach, we designed from two perspectives:
>
> 1. Although the reward functions are calculated differently for the two reasoning approaches, the final reward values are kept the same, so the model is not biased toward any reasoning method merely to obtain higher rewards.
>
> 2. Considering that the simple reasoning step reward is relatively easy to obtain (since it does not enforce generating the corresponding SQL component for subtasks), we bind the reward for complexity analysis to prevent reward hacking. If the model always classifies queries as simple to gain the simple reasoning step reward, it will lose the reward from the complexity analysis.
>
> [1] Guo, Daya, et al. "Deepseek-r1: Incentivizing reasoning capability in llms via reinforcement learning." arXiv preprint arXiv:2501.12948 (2025).
>
> ## Response to Question 2
>
> Sorry for the confusion. Our component pre-generation reward does not increase continuously with the number of components generated. The reward is defined as min(1, |C|/n), where n represents the minimum expected number of SQL components the model should generate, and |C| is the actual number of generated components. Due to the min function, when the number of generated components is less than n, the reward is |C|/n (which is less than 1). When the number of generated components is equal to or greater than n, the maximum reward the model can receive is capped at 1. Thus, this reward is limited and bounded.
>
> The motivation behind this design is to encourage the model to generate at least n SQL components by setting n as a target, while the min function prevents the reward from increasing indefinitely as more components are generated, thereby avoiding the incentive to produce unnecessary or redundant components.
>
>
> ## Response to Question 3
> SPIDER-SYN and SPIDER-REAL are both designed to test model robustness through textual attacks. Both benchmarks are derived from the SPIDER-DEV set.
>
> SPIDER-SYN involves synonym replacement in the natural language queries from SPIDER-dev. For example, a query like “what is the type of the **document** named 'David CV'” is changed to “what is the type of the **file** named 'David CV'” in SPIDER-SYN. This approach breaks the direct correspondence between the natural language query and the database schema fields, thereby attacking the schema linking and challenging the model’s robustness.
>
> In SPIDER-REAL, beyond replacing or removing column names, phrases in the natural language queries are also altered. For instance, “How many pets have a **greater weight than 10**” is changed to “How many pets are **over 10 lbs?**”
>
> We chose these two benchmarks to validate our model’s robustness because they introduce realistic variations and perturbations in the queries that test the model’s ability to generalize under such challenging conditions.
>
> ## Response to Question 4
> Thank you again for your valuable suggestion. We will adjust the placement of Appendices A and B to appear after the main references in the updated version of the paper.

---

> > ### Comment · Reviewer_H8Vy · 2025-11-24
> > **Reply to the authors**
> >
> > I would like to thank the authors for their detailed and sincere rebuttal regarding my concerns and questions. Most of my concerns have been addressed, and the response particularly helped me better understand the mechanism of the pre-generation reward, which I had previously misunderstood.
> >
> > Below are my follow-up comments and remaining questions:
> >
> > Clarification on Weakness 1: to clarify weakness 1, by 'extremely complex query,' I meant 'multi-level nested queries.' My expectation for the limitation discussion was to address whether it is possible to support multi-level pre-generations for handling these types of queries.
> >
> > Follow-up on Question 1: You mentioned that the complex analysis reward is bound to the simple reasoning step reward (which is easier to obtain) to prevent reward hacking. Could you elaborate on the specific mechanism of this binding?

---

> ### Author Response · Authors · 2025-11-24
>
> Thank you very much for your response. I will address each of your questions one by one.
>
>
> ## Response to Performance Analysis of Multi-Level Nested Queries (PART ONE)
> I'm very sorry for misunderstanding your point. Our method indeed has a certain capability for pre-generating components for multi-level nested structures. We take the example of an extremely complex SQL query with parallel and three-level nesting. We still compare the SQL component pre-generation reasoning paradigm and the CoT reasoning paradigm.
>
> **User's Question:**
> Find the names of schools whose enrollment (K-12) is higher than the average enrollment (K-12) of schools in the district that has the highest average SAT Math score.
>
> **1. SQL Component Pre-Generation Reasoning Paradigm:**
> Let's first look at the generation results of the SQL component pre-generation reasoning paradigm:
>
> **Complexity Analysis:**
> The question requires multiple steps: calculating the average SAT Math score by district, finding the district with the highest average SAT Math score, calculating the average enrollment for schools in that district, and then finding the names of schools whose enrollment is higher than this average. This involves multiple joins, aggregations, and subqueries. Therefore, the overall complexity of the question is **challenging**.
>
> **SQL Generation:**
> To find the names of schools whose enrollment (K-12) is higher than the average enrollment (K-12) of schools in the district that has the highest average SAT Math score, we need to follow these steps:
>
> **Step-by-step SQL decomposition:**
> - **Step 1:** Calculate the average SAT Math score by district.
> ```sql
> SELECT s.District, AVG(ss.AvgScrMath) AS avg_math_score
> FROM schools s
> JOIN satscores ss ON s.CDSCode = ss.cds
> GROUP BY s.District
> ```
>
> - **Step 2:** Find the district with the highest average SAT Math score.
> ```sql
> SELECT s.District
> FROM schools s
> JOIN satscores ss ON s.CDSCode = ss.cds
> GROUP BY s.District
> ORDER BY AVG(ss.AvgScrMath) DESC
> LIMIT 1
> ```
>
> - **Step 3:** Calculate the average enrollment for schools in the district with the highest average SAT Math score.
> ```sql
> SELECT AVG(f."Enrollment (K-12)") AS avg_enrollment
> FROM frpm f
> JOIN schools s ON f.CDSCode = s.CDSCode
> WHERE s.District = (
>     SELECT s.District
>     FROM schools s
>     JOIN satscores ss ON s.CDSCode = ss.cds
>     GROUP BY s.District
>     ORDER BY AVG(ss.AvgScrMath) DESC
>     LIMIT 1
> )
> ```
>
> **Final SQL statement:**
> ```sql
> SELECT s.`School Name`
> FROM frpm f
> JOIN schools s ON f.CDSCode = s.CDSCode
> WHERE s.District = (
>     SELECT s.District
>     FROM schools s
>     JOIN satscores ss ON s.CDSCode = ss.cds
>     GROUP BY s.District
>     ORDER BY AVG(ss.AvgScrMath) DESC
>     LIMIT 1
> )
> AND f."Enrollment (K-12)" > (
>     SELECT AVG(f."Enrollment (K-12)")
>     FROM frpm f
>     JOIN schools s ON f.CDSCode = s.CDSCode
>     WHERE s.District = (
>         SELECT s.District
>         FROM schools s
>         JOIN satscores ss ON s.CDSCode = ss.cds
>         GROUP BY s.District
>         ORDER BY AVG(ss.AvgScrMath) DESC
>         LIMIT 1
>     )
> )
> ```
>
> **Case Analysis:**
> From the generation process above, it is clear that step 2 serves as a pre-generated sub-SQL for step 3, and step 3 is a pre-generated sub-SQL for the final SQL. This indicates that our method has a certain supporting effect for multi-level nesting. Additionally, step 1 is also a sub-SQL of the final SQL, and it is parallel in structure with step 3. This shows that our method effectively pre-generates sub-SQL in parallel structures as well.

---

> > ### Author Response · Authors · 2025-11-24
> >
> > ## Response to Performance Analysis of Multi-Level Nested Queries (PART TWO)
> > **2. CoT Reasoning Paradigm:**
> > Let’s also look at the results obtained from the CoT reasoning paradigm to further illustrate the effectiveness of SQL component pre-generation:
> >
> > **Here's a step-by-step breakdown:**
> > 1. **Identify the Tables and Columns:** We need data from the frpm table for enrollment and the satscores table for SAT Math scores.
> > 2. **Join the Tables:** We need to join the frpm table with the satscores table on the CDSCode column to combine enrollment data with SAT scores.
> > 3. **Calculate the Average Enrollment:** We need to calculate the average enrollment (K-12) for each district based on the satscores table.
> > 4. **Order and Limit:** We need to order the results by the average SAT Math score in descending order to find the district with the highest average score and then limit the results to that district.
> > 5. **Filter by Enrollment:** Finally, we filter the schools in that district to find those with enrollment (K-12) higher than the average enrollment of that district.
> >
> > **Here's the SQL query that corresponds to these steps:**
> > ```sql
> > SELECT T1.School
> > FROM frpm AS T1
> > INNER JOIN satscores AS T2 ON T1.CDSCode = T2.cds
> > WHERE T1.`Enrollment (K-12)` > (
> >     SELECT AVG(T1.`Enrollment (K-12)`)
> >     FROM frpm AS T1
> >     INNER JOIN satscores AS T2 ON T1.CDSCode = T2.cds
> >     WHERE T1.`Enrollment (K-12)` > (
> >         SELECT AVG(Enroll12)
> >         FROM satscores
> >     )
> > )
> > ```
> >
> > **Case Analysis:**
> > Taking the third step as an example, the model successfully decomposed the task of calculating the average enrollment (K-12) for each district based on the satscores table. However, there was an error in the SQL generated for this step in the final SQL (the subtask mentioned calculating the average enrollment (K-12) for each district, but the generated final SQL did not reflect this).
> >
> > Through these two cases, on the one hand, we demonstrate that our method has a certain effectiveness for multi-level nested structures; on the other hand, it also highlights that performing SQL pre-generation during task decomposition can enhance the accuracy of the final SQL to some extent, thus affirming the effectiveness of SQL component pre-generation.
> >
> >
> >
> > ## Elaboration on Complexity Analysis Reward and SQL Components Pre-Generation Reward
> >
> > As introduced in section 2.2 of the paper, our reasoning step reward consists of two parts: complexity analysis reward and SQL components pre-generation reward.
> >
> > 1. **Complexity Analysis Reward:** The model must correctly analyze the complexity of the query to earn this portion of the reward.
> >
> > 2. **SQL Components Pre-Generation Reward:**
> >    - For simple queries, the model only needs to decompose the query to obtain this reward (which aligns with the standard CoT method), without needing to perform SQL components pre-generation.
> >    - For moderate and challenging queries, the SQL component pre-generation reward requires both the task decomposition and the pre-generated SQL to be present in order to earn this reward.
> >
> > If the model attempts to "cut corners" by predicting all queries as simple (thus only needing to decompose the query to earn the SQL component pre-generation reward), it will result in many incorrect complexity predictions, leading to the loss of the complexity analysis reward. On the other hand, if the model correctly analyzes the complexity and performs component pre-generation, it can earn both the complexity analysis reward and the SQL components pre-generation reward.
> >
> > This mechanism effectively prevents reward hacking.
> >
> > Thank you once again for your feedback. If you have any further questions, feel free to ask, and I will continue to provide detailed answers to each one!

---

> > > ### Comment · Reviewer_H8Vy · 2025-11-25
> > >
> > > I thank the authors for their detailed responses to my additional questions. Their answers have fully addressed my inquiries.

---

> > > > ### Author Response · Authors · 2025-11-25
> > > >
> > > > Dear Reviewer,
> > > >
> > > > Thank you very much for your thoughtful and constructive comments. We sincerely appreciate the time and effort you have taken to review our work and provide valuable feedback. We are glad that our responses have addressed your inquiries satisfactorily.
> > > >
> > > > We have updated the manuscript based on your suggestions, including the applicability analysis of multi-level nested SQL queries, improvements to the paper’s formatting, style, and overall presentation, as well as additional explanations regarding the relevant datasets and key techniques. We will upload the latest version of the paper to OpenReview as soon as possible.
> > > >
> > > > We look forward to any further suggestions you may have.
> > > >
> > > > Best regards,

---

### Official Review · Reviewer_TWiP · 2025-10-31

**Soundness:** 3
**Presentation:** 3
**Contribution:** 3
**Rating:** 6
**Confidence:** 4

**Summary:**

This paper identifies a clear and intuitive problem with existing Text-to-SQL models and proposes a novel, multi-part solution to address it. The core idea is to change the model's reasoning process to first pre-generate individual SQL components before assembling the final query, thereby solving the "semantic blind spot". This is trained using a sophisticated reinforcement learning strategy featuring a novel AST-based reward

**Strengths:**

1. The paper's core premise, the "semantic blind spot", is a significant and well-articulated insight. The authors correctly observe that the standard, token-by-token generation process is unlike how humans write complex SQL. The analogy of a human writing an inner query first and then building the outer query around it is a powerful justification for their approach.
2. The proposed "SQL components pre-generation" reasoning paradigm  is a logical and creative solution to the identified problem. By having the model first generate the independent semantic components, it effectively provides the necessary context for the model to use when constructing the final, complex query, mitigating the blind spot.

**Weaknesses:**

1. The "SQL Components Pre-generation Reward" is based on generating a minimum number of components (e.g., 2 for moderate, 3 for challenging). This rule-based reward could potentially be "hacked" by the model, which might learn to generate multiple incorrect or useless components just to satisfy the count and receive the reward. While the AST matching reward would counteract this by giving low scores to bad components, the pre-generation reward itself is not directly tied to the quality or utility of the components.
2. The performance of the complexity analysis module feels like a potential weak link. The F1 scores for prediction (0.77, 0.75, 0.87)  are not exceptionally high. The paper argues this is a good thing, suggesting the model learns a more nuanced complexity "beyond just component count". However, this could also be a post-hoc justification. An error in this initial step (e.g., misclassifying a complex query as "simple") would cause the model to skip the pre-generation step entirely, leading to the very "semantic blind spot" problem this paper aims to solve.
3. The authors acknowledge this in their conclusion. While AST matching is superior to n-gram matching, it is not a complete solution for semantic equivalence. Two SQL queries can be logically identical (e.g., a JOIN vs. a correlated subquery in the WHERE clause) but have completely different ASTs. The current normalization (removing aliases, sorting AND nodes)  is a good first step, but this reward mechanism may still unfairly penalize valid, alternative SQL queries.

**Questions:**

1. The proposed reasoning paradigm is a multi-step process, which will inherently increase inference latency and computational cost compared to a single-pass generation. The paper critiques another method (Rex-SQL) for its high cost  but does not provide any analysis of its own method's overhead, which is a key practical consideration, especially in RL training.

---

> ### Author Response · Authors · 2025-11-21
>
> Thank you for your detailed and insightful review of our paper. Below, we address each point in detail:
> ## Response to Weakness 1
> Your concern is very valid. We have taken the following approaches to reduce the interference caused by generating redundant components:
>
> a. First, we estimate the number of SQL components in the training set using a LLM and obtain the average number of components for the moderate and challenging categories. We then set the expected number of generated components to be less than the average number, using this conservative approach to avoid generating redundant components.
>
> b. We apply a clipping function min(1, |c|/n) to the reward value, where c is the number of generated components and n is the minimum expected number of components. This limits the reward value so that it does not increase with the number of generated components beyond the expected minimum.
>
> c. Finally, as you suggested, we use AST matching to assign low scores to the whole when there are many redundant components.
>
> We demonstrate with a set of data that this approach alleviates the redundancy of components to a certain extent.
>
> |            | simple  | moderate | challenging |
> |------------|---------|----------|-------------|
> | **simple**      | 22.22%  | 44.87%   | 57.14%      |
> | **moderate**    | 52.14%  | 37.62%   | 34.92%      |
> | **challenging** | 62.26%  | 48.15%   | 47.43%      |
>
>
> This table is the confusion matrix of our complexity analysis results, where the row labels represent the true labels and the column labels represent the predicted labels. The values in the table indicate the SQL error rates.
>
> a. From the first row of simple data, we can see that when the simple label is misclassified as moderate or challenging, component pre-generation occurs, and the SQL error rate significantly increases. This shows that generating redundant components indeed leads to a decrease in SQL accuracy.
>
> b. From the second row of moderate data, when moderate is misclassified as challenging, more SQL components are generated. At this time, the SQL error rate decreases from 37.62% to 34.92%, indicating the conservativeness of our setting for the number of components to be pre-generated.
>
> From the above analysis, it can be seen that the specific number of pre-generated components has a significant impact on SQL generation accuracy. On one hand, this demonstrates the effectiveness of our pre-generation method; on the other hand, it also suggests there is room for further improvement in reasonably setting the number of pre-generated components.
>
>
> ## Response to Weakness 2
> Yes, this issue does exist. As shown by the data in the table in Weakness 1, when moderate and challenging queries are misclassified as simple, the SQL generation error rate increases. This further demonstrates the effectiveness of the component pre-generation and also indicates that the complexity analysis module provides valuable guidance for SQL generation accuracy. In future work, we will further explore more reasonable methods for complexity analysis.
>
> ## Response to Weakness 3
> Yes, it is impossible to cover every valid SQL representation through rule-based methods. Therefore, we use maximum common subtree matching to provide rewards at the SQL component granularity. This finer-grained reward mechanism reduces the interference caused by SQL diversity, because the smaller the granularity, the fewer the possible variations, which helps to mitigate unfairness in reward values.
>
> Although this does not fundamentally eliminate the existence of such unfairness, it greatly improves the fairness of the reward mechanism. Finally, our experiments show that this approach effectively ensures that positive samples receive higher reward values than negative samples, as demonstrated in Table 4 of the paper.

---

> ### Author Response · Authors · 2025-11-21
>
> ## Response to Weakness Quetsion 1
>
> Thank you for your feedback. Although our method introduces steps such as complexity analysis and component pre-generation, we still use single-round training and inference, so the token consumption is similar to that of the standard GRPO algorithm. The training method used in Rex-SQL is multi-round interactive verification, and since Rex-SQL does not provide analysis on training and inference latency in their paper, we are unable to make a direct comparison.
>
> We have supplemented the comparison of various token consumptions. As shown in the table, compared to the standard GRPO training method (which executes both accuracy and format rewards), our method increases token consumption by only 0.23k, but improves execution accuracy by 1.89%. Compared to the MCTS-based method such as Alpha-SQL, our token consumption and inference latency have significant advantages.
>
> | Methods    | Model                 | EX    | Input tokens (k) | Output tokens (k) | Total tokens (k) | Training Latency (s/step) | Inference Latency (s/question) |
> |------------|----------------------|-------|------------------|-------------------|------------------|---------------------------|-------------------------------|
> | Alpha-SQL  | Qwen2.5-Coder-7B     | 66.8  | 138              | 72.2              | 200.2            | -                         | 377.1                        |
> | GRPO       |  Qwen2.5-Coder-7B    | 63.89 | 0.38             | 0.29              | 0.67             | 250.82                    | 3.05                         |
> | Ours       |    Qwen2.5-Coder-7B  | 65.78 | 0.42             | 0.48              | 0.90             | 267.73                    | 5.08                         |

---

> > ### Comment · Reviewer_TWiP · 2025-11-25
> >
> > Thanks for the authors’ detailed responses and our discussion will surely be valuable for refining this paper, and I believe the current score is appropriate.

---

> > > ### Author Response · Authors · 2025-11-25
> > >
> > > Dear Reviewer,
> > >
> > > Thank you very much for your thoughtful comments and for recognizing the value of our responses and discussion. We have updated the manuscript to incorporate the suggestions you mentioned, including the discussion on computational resource consumption for training and inference, the limitations of  the AST matching reward mechanism and the SQL component pre-generation approach. These suggestions have made our work more complete and rigorous. We appreciate your evaluation and believe that your feedback will help us further improve the paper. We will continue to work hard to refine our work and will upload the latest version of the manuscript to the OpenReview platform as soon as possible.
> > >
> > > Best regards,

---

### Official Review · Reviewer_mVHC · 2025-11-01

**Soundness:** 3
**Presentation:** 3
**Contribution:** 2
**Rating:** 4
**Confidence:** 3

**Summary:**

This paper addresses a key limitation in LLMs for complex Text-to-SQL tasks, which the authors term the "semantic blind spot." This problem arises because LLMs generate SQL queries sequentially, token-by-token, making them unable to utilize information from parts of the query that have not yet been generated. To address this, the authors propose a two-stage reasoning approach: the model first analyzes query complexity and, for complex cases, pre-generates independent SQL components. These components then inform the second stage, where the final SQL query is constructed. Training uses reinforcement learning with an AST–based reward system that scores structural correctness via maximum connected subtree matching, offering greater robustness than text-based evaluation.

**Strengths:**

S1: The paper presents a novel viewpoint on a familiar issue in sequential generation by conceptualizing it as a “semantic blind spot.” Although breaking tasks into smaller components is not a new idea, the particular approach of first generating independent SQL elements and then combining them—guided by an advanced AST-based reward system—represents a creative and revitalized direction within the Text-to-SQL field.

S2: The methodological approach is sound and well-justified. The proposed two-stage reasoning process, beginning with complexity analysis and proceeding to component pre-generation, directly addresses the identified problem. The use of RL with a custom AST-matching reward is a robust choice for guiding the model to produce structurally correct and independently meaningful SQL fragments.

S3: The experimental design is rigorous, particularly the demonstration of cross-dataset generalization (training on BIRD, evaluating on SPIDER), which speaks to the robustness of the learned strategies rather than mere dataset overfitting. The detailed ablation studies further enhance the quality by systematically isolating the contributions of different components of the proposed method.

**Weaknesses:**

W1:
The paper introduces a “semantic blind spot” and proposes “SQL component pre-generation” to address it. However, its approach—decomposing queries into smaller parts and recombining them—closely parallels existing Chain-of-Thought and task decomposition methods. The distinction between the proposed method and prior intermediate-step generation strategies remains unclear.

W2:
The paper reports strong generalization by training on BIRD and testing on SPIDER. This may stem from BIRD’s inherent complexity rather than the method itself. Stronger evidence would come from reverse generalization—training on SPIDER and testing on BIRD—to assess transferability from simpler datasets.

W3:
While effective, the method adds costly training steps: complexity analysis, SQL pre-generation, and AST-based reward computation. The paper lacks analysis of the additional computational overhead, such as training time, GPU consumption, or inference latency.

**Questions:**

Q1: The entire "pre-generation" strategy is predicated on the initial "complexity analysis" step, which determines if a query is "simple," "moderate," or "challenging." Table 4 reveals that the F1 scores for this classification are modest (ranging from 0.77 to 0.87). This implies that a significant number of queries will be misclassified. Can the authors provide a detailed error analysis of this module?

---

> ### Author Response · Authors · 2025-11-21
>
> Thank you for your detailed and insightful review of our paper. Below, we address each point in detail:
> ## Response to Weakness 1: Comparison of SQL Component Pre-generation and CoT Reasoning Paradigms (PART Ⅰ)
> ### 1. Explanation of Differences from CoT
> In Figures 1 and 5 of our paper, we illustrate the differences between the standard Chain-of-Thought (CoT) paradigm and the SQL Components Pre-generation reasoning paradigm. Using Figure 5 as an example, we further explain the distinctions and advantages of our approach compared to CoT, supported by experimental results, to help clarify our ideas.
>
> Our method differs from CoT in two main aspects:
>
> 1. **Different Objectives:** Our method aims to enable the model to utilize the semantic information of SQL components to be generated when producing the final SQL, addressing the semantic blind spot of pending components in the Text-to-SQL task. In contrast, CoT’s goal is only to reduce task complexity through decomposition and does not consider the semantic blind spot problem of pending components during final SQL generation. Although CoT performs well in general domains, challenges remain when applying it to Text-to-SQL tasks.
>
> 2. **Different Reasoning Paradigms:**
> To better explain, we first define two concepts:
> - **Task Unit:** A subtask obtained by decomposing the question.
> - **SQL Unit:** The SQL component that solves the corresponding task unit.
>
>
> Consider the user query:
> *"What number of patients with a degree of thrombosis level 2 and ANA pattern of only S, have a level of anti-Cardiolipin antibody (IgM) 20% higher than average?"*
>
> **Chain-of-Thought Reasoning Paradigm:**
> - Task Unit 1: Calculate the average aCL IgM level. We need to calculate the average aCL IgM level from the Examination table.
> - Task Unit 2: Determine the threshold for 20% higher than average. We need to calculate 20% higher than the average aCL IgM level.
> - ...
> Final SQL:
> ```sql
> SELECT COUNT(*) AS number_of_patients
> FROM Examination E
> WHERE E.Thrombosis = 2
>   AND E.ANA_Pattern = 'S'
>   AND E.aCL_IgM > (SELECT AVG(aCL_IgM) * 1.2 FROM Examination);
> ```
>
> **SQL Components Pre-generation Reasoning Paradigm:**
> - Task Unit 1: Filter patients based on thrombosis level and ANA pattern.
> - SQL Unit 1: `WHERE Thrombosis = 2 AND ANA_Pattern = 'S'`
> - Task Unit 2: Determine the threshold for aCL IgM.
> - SQL Unit 2: `SELECT AVG(aCL_IgM) * 1.2 AS threshold FROM Examination WHERE Thrombosis = 2 AND ANA_Pattern = 'S'`
> - ...
> Final SQL:
> ```sql
> SELECT COUNT(T1.ID) AS count_patients
> FROM Examination AS T1
> WHERE T1.Thrombosis = 2 AND T1.ANA_Pattern = 'S'
>   AND T1.aCL_IgM > (SELECT AVG(aCL_IgM) * 1.2
>                     FROM Examination
>                     WHERE Thrombosis = 2 AND ANA_Pattern = 'S')
> ```
>
> It can be observed that the CoT decomposition typically produces only task units without generating the corresponding SQL units to solve those tasks. The disadvantage of this approach is that when generating the final SQL statement, the model only has access to the semantic information of the task units, while all SQL components remain invisible. Since task units are usually expressed in natural language, there exists a semantic gap between these natural language task units and the structured SQL units. This gap forces the model, when generating the final SQL, to both infer the SQL units corresponding to each task unit and consider how to compose these SQL units correctly, increasing the complexity of the task.
>
> This challenge is particularly pronounced for complex SQL queries. Even if the task units are correctly decomposed, the semantic gap between natural language and SQL units often leads to difficulties in correctly converting all task units into accurate SQL units and combining them appropriately.
>
> For example, in this case, the CoT paradigm correctly decomposes the task “We need to calculate 20% higher than the average aCL IgM level.” However, the final SQL generated from all task units contains errors.
>
> Our approach further optimizes the reasoning process specifically for the Text-to-SQL task. While decomposing the question into task units, it simultaneously generates the corresponding SQL units that solve each task unit. Therefore, when generating the final SQL, the model can leverage the semantic information of already generated SQL units. This strategy bridges the gap between natural language task units and structured SQL statements. Consequently, the model only needs to consider how to combine different SQL units when generating the final SQL, without having to infer each task unit’s SQL counterpart at that stage. Compared with the CoT paradigm, this reduces the complexity of the Text-to-SQL task even further.

---

> ### Author Response · Authors · 2025-11-21
>
> ## Response to Weakness 1: Comparison of SQL Component Pre-generation and CoT Reasoning Paradigms (PART Ⅱ)
>
> We formalize the two task paradigms as follows:
>
> - **CoT formalization:**
>   Given the generated sequence $(t_1, t_2, t_3, ..., t_n)$, the sequence to be generated is the final SQL $(s_1, s_2, ..., s_n)$.
>   The final SQL generation can be represented as:
>
>   $$
>    SQL = (s_1, s_2, ..., s_n) = f(t_1, t_2, ..., t_n)
>   $$
>
> - **Our method:**
>   Given the generated sequence $(t_1, s_1, t_2, s_2, ..., t_n, s_n)$, the sequence to be generated is the final SQL $(s_1, s_2, ..., s_n)$.
>   The final SQL generation is:
>
>   $$
>   SQL = (s_1, s_2, ..., s_n) = f(t_1, s_1, t_2, s_2, ..., t_n, s_n)
>   $$
> where `f` denotes LLMs, `t` denotes sub-tasks, and `s` denotes SQL components.
>
>
> ### 2. Experimental Validation
>
> We further validated our method by comparing the two reasoning paradigms on the bird-dev dataset of DeepSeek-v3.2 through prompting. The results demonstrate that the SQL components pre-generation reasoning paradigm significantly outperforms the CoT paradigm in the Text-to-SQL task, summarized as follows:
>
> | Models         | Methods                     | Simple | Moderate | Challenging | Total |
> |----------------|-----------------------------|--------|----------|-------------|-------|
> | DeepSeek-v3.2  | CoT                         | 62.16  | 42.73    | 36.11       | 53.72 |
> | DeepSeek-v3.2  | SQL Components Pre-generation | **62.92**  | **42.80**    | **39.58**       | **54.63** |
>
> `(Execution Accuracy Comparison between Our Method and CoT on BIRD-DEV)`
>
>
> ## Response to Weakness 2: Training on Spider and Validating on BIRD
> You are absolutely right. We have corresponding experimental settings in the paper. In the generalization experiments (Figure 4(a)), we demonstrate the performance of a model trained on Spider and evaluated on Bird. The baseline GRPO reward function achieves an execution accuracy of 61.60%, which increases to 62.78% after incorporating our designed reward function.
>
> ## Response to Weakness 3: Lack of Efficiency Analysis
> While our method introduces complexity analysis and SQL components pre-generation steps, we still use single-round training and inference. Therefore, the overhead is comparable to the standard GRPO method. We provided comparisons of various consumption metrics. As shown, compared to the standard GRPO training method (execution accuracy reward and format reward), our method only increases token consumption by 0.23k but improves execution accuracy by 1.89. Compared to MCTS-based methods such as Alpha-SQL, our computational cost and inference latency are significantly lower.
>
> | Methods    | Model              | EX    | Input tokens (k) | Output tokens (k) | Total tokens (k) | Training Latency (s/step) | Inference Latency (s/question) |
> |------------|--------------------|-------|------------------|------------------|------------------|---------------------------|-------------------------------|
> | Alpha-SQL  | Qwen2.5-Coder-7B   | 66.8  | 138              | 72.2             | 200.2            | -                         | 377.1                         |
> | GRPO       |   Qwen2.5-Coder-7B    | 63.89 | 0.38             | 0.29             | 0.67             | 250.82                    | 3.05                          |
> | **Ours**       |   Qwen2.5-Coder-7B     | 65.78 | 0.42             | 0.48             | 0.90             | 267.73                    | 5.08                          |
>
> `(Efficiency Analysis)`

---

> ### Author Response · Authors · 2025-11-25
>
> ## Response to Question 1: Error Analysis of Complexity Analysis Module
>
>
> **Response:**
>
> We have conducted a further error analysis of the complexity analysis module.
>
> In Table 1, we summarize the confusion matrix for complexity prediction, with rows representing the true complexity labels and columns representing the predicted complexity categories.
>
> |            | simple | moderate | challenging |
> |------------|--------|----------|-------------|
> | **simple**      | 603    | 78       | 7           |
> | **moderate**    | 69     | 388      | 63          |
> | **challenging** | 27     | 53       | 245         |
> `(Table 1: Confusion Matrix for Complexity Prediction on BIRD-DEV)`
>
> In Table 2, we present the SQL error rates for each predicted category, with rows showing the true complexity labels and columns showing the predicted complexity categories.
> |            | simple  | moderate | challenging |
> |------------|---------|----------|-------------|
> | **simple**      | 22.22%  | 44.87%   | 57.14%      |
> | **moderate**    | 52.14%  | 37.62%   | 34.92%      |
> | **challenging** | 62.26%  | 48.15%   | 47.43%      |
>
> `(Table 2: SQL Error Rates by Predicted Category on BIRD-DEV)`
>
> Our key findings are as follows:
>
> 1. **SQL component pre-generation is effective for moderate and challenging questions.** For example, as shown in Table 2, when the true complexity label is `moderate` but predicted as `simple`, the error rate is 52.14%. However, if predicted as `challenging`, the error rate decreases to 34.92%. Similarly, when the true complexity label is `challenging` but misclassified as `simple`, the error rate reaches 62.26%, while classification as `moderate` reduces the error rate to 48.15%. This further demonstrates the effectiveness of SQL component pre-generation for `moderate` and `challenging` SQL queries. This is because when `moderate` or `challenging` questions are misclassified as `simple`, SQL component pre-generation is not performed, leading to poorer performance. However, when `moderate` questions are misclassified as `challenging`, or `challenging` questions as `moderate`, SQL component pre-generation is still applied, resulting in lower error rates.
>
> 2. **Not all queries require component pre-generation.** When `simple` queries are misclassified as `moderate` or `challenging`, the SQL component pre-generation inference paradigm is applied, resulting in increased SQL generation error rates of 44.87% and 57.14%, respectively. This observation is consistent with the experimental results reported in Table 3 of the paper.
>
> 3. **Misclassification often occurs between adjacent complexity levels.** As shown in Table 1, for `simple` queries, there are 78 cases misclassified as `moderate`, while only 7 cases are misclassified as `challenging`. For `moderate` queries, misclassifications are more evenly distributed between `simple` and `challenging` labels. A similar pattern is observed for `challenging` queries. This suggests that errors in complexity classification mainly arise from the ambiguous boundaries between complexity levels. Therefore, a more reasonable definition and analysis method for complexity warrants further exploration in future work.

---

### Official Review · Reviewer_jJZB · 2025-11-01

**Soundness:** 2
**Presentation:** 3
**Contribution:** 3
**Rating:** 4
**Confidence:** 4

**Summary:**

This paper addresses the challenge faced by large language models (LLMs) in generating complex SQL queries from natural language, specifically identifying a “semantic blind spot” problem: during sequential generation, LLMs have limited access to semantic information pertaining to pending SQL components, making it hard to accurately assemble complex queries. To mitigate this, the authors propose a complexity-guided SQL components pre-generation paradigm, whereby the model first generates independent semantic SQL components explicitly before rendering the final SQL query. This process is guided via reinforcement learning, employing the GRPO framework with a bespoke reward structure. Central to the approach is a maximum connected subtree matching reward mechanism based on SQL Abstract Syntax Trees (ASTs), which offers structurally informed, fine-grained feedback.

**Strengths:**

1. The paper is clearly written, and the figures are easy to understand.

2. The discussion on semantic blind spots in pending SQL components is meaningful and highlights a real limitation in current LLM-based SQL generation.

3. Compared to existing methods focusing on improving LLM SQL reasoning, the proposed maximum connected subtree matching reward mechanism is novel.

**Weaknesses:**

1. The main concern is that the proposed method only delivers marginal improvements and is sometimes even weaker than the baselines. As shown in Table 1, the method only achieves the best results on SPIDER-SYN, while on the other four datasets, it underperforms compared to the baselines. Since all baselines use LLMs of the same scale, the lower accuracy calls into question the effectiveness of the proposed framework.

2. The authors claim the method addresses the challenge of generating complex SQL statements, but they do not evaluate on Spider2, which is more complex compared to BIRD and SPIDER. Instead, they mostly use the simpler Spider and its extensions for evaluation. This raises doubts about the method’s ability to handle truly complex SQL queries.

3. Data in real SQL scenarios is different from public datasets and often lacks gold SQL queries. The proposed method depends on gold SQL-based AST construction and reward modeling, which casts doubt on its real-world usability and practicality.

4. The paper lacks a discussion and comparison with a relevant baseline: Alpha-SQL: Zero-Shot Text-to-SQL using Monte Carlo Tree Search, which uses MCTS to improve the SQL reasoning abilities of local LLMs.

**Questions:**

Please check the Weaknesses.

---

> ### Author Response · Authors · 2025-11-21
>
> Thank you for your detailed and insightful review of our paper. Below, we address each point in detail:
> ## Weakness1: Advantages Compared to Baselines
> We selected the best-performing baselines on the Spider and BIRD datasets respectively for further comparison to highlight the advantages of our method:
>
> 1. **Better Generalization and Robustness:**
> Most baseline methods perform well on only one of the datasets, either BIRD or Spider. For example, SQL-R1 achieves its best results on the Spider dataset, with a maximum execution accuracy (EX) of 87.6 on Spider-test. However, its EX drops to 60.8 on BIRD, and to 76.7 and 76.8 on Spider variants (Spider-Syn and Spider-Real), respectively. In contrast, our method outperforms SQL-R1 by 5% on the realistic and complex BIRD dataset, leads by 3–4% on the Spider variant datasets, and still maintains a high accuracy above 83% on the original Spider dataset. These results demonstrate that our approach possesses stronger generalization capabilities and robustness.
>
> |           | bird-dev | spider-dev | spider-test | spider-syn | spider-real |
> |-----------|----------|------------|-------------|------------|-------------|
> | SQL-R1-7B | 60.8     | **83.8**   | **87.6**    | 76.7       | 76.8        |
> | ours      | **65.8** | 83.0       | 84.4        | **80.7**   | **79.9**    |
> `(Better Generalization and Robustness)`
>
> 2. **Lower Resource Consumption:**
> Taking Reward-SQL-7B[2] as an example, since it is the only method that surpasses ours on the bird dataset. The comparison results are shown in the table below. It can be seen that reward-sql used 18,105 cold start data, 31,635 reward model training data, and 9,428 bird training data. In addition, reward-sql also trained cold start and reward models, etc. In contrast, we only used 9,428 bird training data. On the bird dataset, our method is only 0.6% lower than reward-sql, and on the spider datasets, our method outperforms reward-sql.
>
> | Method         | Training Data Distribution                             | Model Used            | bird-dev | spider-dev | spider-test | spider-syn | spider-real |
> |----------------|-------------------------------------------------------|----------------------|----------|------------|-------------|------------|-------------|
> | Reward-sql-7B  | 18,105 cold start data; 31,636 reward model training data; 9,428 bird-train data | Cold start model; Reward model; GRPO model | **66.4**     | 77.0       | 78.5        | 78.1       | 75.4        |
> | **ours**           | 9,428 bird-train data                                  | GRPO model           | 65.8     | **83.0**       | **84.4**        | **80.7**       | **79.9**        |
>
> `(Lower Resource Consumption)`
>
> **As a side note**, We originally planned a more fine-grained comparison between our method and Reward-SQL on `challenging` data from the BIRD benchmark. However, since the Reward-SQL paper does not report execution accuracy across questions of varying complexity, we reproduced their greedy search results using the provided code and model checkpoint. The reproduced results are as follows. To ensure rigor, we evaluated five runs with different inference parameters (temperature values of 0, 0.2, 0.4, 0.6, and 0.8) and selected the best execution accuracy among them for display in the table below. The best accuracy was achieved at temperature 0 (EX = 62.32), and the worst at temperature 0.4 (EX = 61.73), indicating that parameter variation has minimal impact on their model.
>
> | Method      | Simple | Moderate | Challenging | Total |
> |-------------|--------|----------|-------------|-------|
> | Reward-SQL  | -      | -        | -           | **66.4** |
> | Reward-SQL* | 68.11  | 53.98    | 52.08       | 62.32 |
> | ours        | 70.38  | 57.42    | **63.19**   | 65.78 |
>
> `(* indicates our reproduced results)`
>
> The Reward-SQL execution accuracy reported in our paper is directly cited from their original publication. We cannot completely rule out potential reproduction errors; however, if no such errors exist, this not only confirms the advantages of our approach described earlier but also demonstrates that, when compared to baselines of similar parameter scale, our method achieves the highest execution accuracy on the BIRD-dev dataset.
>
> [1] Ma, Peixian, et al. "Sql-r1: Training natural language to sql reasoning model by reinforcement learning." arXiv preprint arXiv:2504.08600 (2025).
>
> [2] Zhang, Yuxin, et al. "Reward-sql: Boosting text-to-sql via stepwise reasoning and process-supervised rewards." arXiv preprint arXiv:2505.04671 (2025).

---

> ### Author Response · Authors · 2025-11-21
>
> ## Weakness2: Lack of Evaluation on Spider2
> | Model               | Reward function | Spider-2.0-SQLite |
> |---------------------|-----------------|-------------------|
> | GPT-4o              | -               | 15.6              |
> | DeepSeek-V3         | -               | 15.6              |
> |       |                |               |
> | Qwen                | -               | 1.5               |
> | Qwen+GRPO           | base rewards    | 5.9               |
> | **Qwen+GRPO (Ours)**   | all rewards     | **8.9**               |
> |       |                |               |
> | Omini-sql-7B        | -               | 10.4              |
> | Omini-sql-7B+GRPO   | base rewards    | 11.1              |
> | **Omini-sql-7B+GRPO (ours)** | all rewards | **14.1**              |
>
> We have now supplemented our experiments on Spider2.0. Specifically, we selected `Qwen2.5-Coder-7B-Instruct` and `OminiSQL-7B` as base models. The experimental results are shown in the table, where `base rewards` refers to the standard GRPO reward design, which includes only `execution result reward` and `format reward`. As shown, the Qwen model achieves an execution accuracy of only 1.5% on Spider2. After applying `base rewards`, the accuracy rises to 5.9%, demonstrating that the GRPO algorithm significantly improves the model's ability to generate SQL on this challenging benchmark. With our proposed `SQL components pre-generation reward` and `maximum connected subtree matching reward` mechanisms (all rewards), the execution accuracy further increases by 2.9% (from 5.9% to 8.8%), verifying that our method remains effective on Spider2.0. Similarly, on OminiSQL-7B, our method improves accuracy by 3% over the standard GRPO algorithm, and the gap with GPT-4o and DeepSeek-V3 is reduced to only 1.5%.
>
> **This demonstrates the ability of our method to generate complex SQL queries and its effectiveness on more challenging datasets such as Spider2.**
>
> **Additionally, a more complex SQL case study on Spider 2.0 is provided in Appendix 4.3, further demonstrating the effectiveness of our method on complex SQL generation tasks.**
>
> ## Weakness3: Real-World Data Lacks Gold SQL
> The use of gold SQL is required only during the training phase when applying the GRPO algorithm, specifically for constructing syntax trees to define the reward function and enhance SQL generation accuracy. However, once the model is fully trained, the inference (or deployment) process does not rely on gold syntax trees at all. In practical usage, the trained model generates SQL queries in the same manner as the base Qwen series models, without requiring gold SQL or AST construction. Therefore, our method remains fully applicable and practical in real-world scenarios where gold SQL queries are unavailable.
>
> ## Weakness4: Missing Comparison with Alpha-SQL
>
> Since our method differs significantly from Alpha-SQL, we did not include a direct comparison with Alpha-SQL in the paper. Alpha-SQL uses Monte Carlo Tree Search (MCTS) to repeatedly invoke the model and search for better SQL generation paths, which results in substantial resource consumption.
>
> Here, we provide a supplementary comparison with Alpha-SQL, as shown in the tables below. It can be seen that our token consumption and inference latency are much lower than Alpha-SQL’s. Although our EX metric is slightly lower than Alpha-SQL’s, Alpha-SQL requires about 200k tokens to solve one question, while our method only consumes about 0.9k tokens. Regarding inference latency, our method takes on average 5.08 seconds per question, whereas Alpha-SQL takes 377.1 seconds. In practical application scenarios, when the EX metric difference is not large, lower cost and faster response are more desirable.
>
> | Methods   | Models           | EX    | Input tokens (k) | Input tokens (k) | Total tokens (k) | Latency (s) / Question |
> |-----------|------------------|-------|------------------|------------------|------------------|-----------------------|
> | Alpha-SQL | Qwen2.5-Coder-7B | 66.8  | 138              | 72.2             | 200.2            | 377.1                 |
> | Ours      |  Qwen2.5-Coder-7B       | 65.78 | 0.42             | 0.48             | 0.90             | 5.08                  |
>
> `(Performance Comparison with Alpha-SQL on BIRD-DEV)`
>
> We also compared performance on question of different complexity levels. On the challenging subset, our method shows certain advantages over Alpha-SQL, as shown below:
>
> |          | simple | moderate | challenging | total  |
> |----------|--------|----------|-------------|--------|
> | Alpha-SQL| 72.6   | 59.3     | 53.1        | 66.8   |
> | Ours     | 70.38  | 57.42    | **63.19**   | 65.8  |
>
> `(Execution Accuracy Comparison with Alpha-SQL on BIRD-DEV)`

---

> ### Author Response · Authors · 2025-11-28
>
> ## Case Analysis on Spider 2.0 (PART Ⅰ)
> To facilitate your review, we have also placed the case analysis on Spider 2.0 from the Appendix 4.3 onto the discussion page. This is an extremely complex SQL example intended to demonstrate the effectiveness of our approach on such highly complex cases in Spider 2.0:
>
> ---
>
> **User's Question:**
> Could you find out the average percentage of the total lifetime sales (LTV) that occur in the first 7 and 30 days after a customer's initial purchase? Also, include the average total lifetime sales (LTV). Please exclude customers with zero lifetime sales. The 7- and 30-day periods should be based on the exact number of hours-minutes-seconds, not calendar days.
>
> ---
>
> `First, let's look at the result obtained using the CoT (Chain-of-Thought) reasoning paradigm:`
>
> ### 1. CoT Reasoning Paradigm
>
> To answer this question, we need to calculate the average percentage of the total lifetime sales (LTV) that occur in the first 7 and 30 days after a customer's initial purchase. We also need to include the average total lifetime sales (LTV) and exclude customers with zero lifetime sales.
>
> Here’s the step-by-step reasoning process:
> 1. Identify the relevant tables: The relevant tables for this query are **payment** and **customer**. The payment table contains the payment information, and the customer table contains customer information.
> 2. Join the tables: Join the payment table with the customer table on the `customer_id` column to link each payment to a specific customer.
> 3. Calculate the total lifetime sales (LTV): Sum up the total sales for each customer.
> 4. Calculate the sales within the first 7 and 30 days: Filter the payments to include only those that occurred within the first 7 and 30 days after the customer's initial purchase.
> 5. Calculate the average percentage: Calculate the average percentage of the total lifetime sales that occur within the first 7 and 30 days.
> 6. Exclude customers with zero lifetime sales.
>
> Here is the SQL query that corresponds to this reasoning process:
>
> ```sql
> SELECT
>   CAST(SUM(CASE WHEN julianday(T1.payment_date) - julianday(T2.create_date) BETWEEN 0 AND 7 THEN T1.amount ELSE 0 END) AS REAL) * 100 / SUM(T1.amount) AS avg_7_day_sales,
>   CAST(SUM(CASE WHEN julianday(T1.payment_date) - julianday(T2.create_date) BETWEEN 0 AND 30 THEN T1.amount ELSE 0 END) AS REAL) * 100 / SUM(T1.amount) AS avg_30_day_sales,
>   SUM(T1.amount) AS avg_ltv
> FROM
>   payment AS T1
> JOIN
>   customer AS T2 ON T1.customer_id = T2.customer_id
> WHERE
>   T1.amount > 0;
> ```
>
> **Case analysis:**
> We can see that the CoT reasoning paradigm reasonably decomposes the complex problem into necessary sub-steps. However, there is a problem when generating the final SQL: for example, in step 3, although the model correctly identifies the **need to calculate the total lifetime sales (LTV), it fails to group the data by customer in the SQL, which is essential.**

---

> ### Author Response · Authors · 2025-11-28
>
> ## Case Analysis on Spider 2.0 (PART Ⅱ)
>
> `Next, let’s examine the results generated by the SQL component pre-generation reasoning paradigm:`
>
> ### 2. SQL Component Pre-generation Reasoning Paradigm (Ours)
>
> **Complexity Analysis:**
> The question requires calculating averages for two different time periods (7 and 30 days) after a customer's initial purchase, along with the total lifetime sales. This involves multiple aggregations and conditional logic. Therefore, the overall complexity of the question is **Challenging**.
>
> **SQL Generation:**
> To calculate the average percentage of the total lifetime sales (LTV) that occur in the first 7 and 30 days after a customer's initial purchase, we follow these steps:
>
> - Step 1: Calculate the total lifetime sales (LTV) for each customer.
>
> ```sql
> SELECT
>     c.customer_id,
>     SUM(p.amount) AS total_ltv
> FROM customer c
> JOIN payment p ON c.customer_id = p.customer_id
> GROUP BY c.customer_id
> ```
>
> - Step 2: Determine the first purchase date for each customer.
>
> ```sql
> SELECT
>     c.customer_id,
>     MIN(p.payment_date) AS first_purchase_date
> FROM customer c
> JOIN payment p ON c.customer_id = p.customer_id
> GROUP BY c.customer_id
> ```
>
> - Step 3: Calculate the sales in the first 7 and 30 days after the first purchase.
>
> ```sql
> SELECT
>     c.customer_id,
>     SUM(CASE WHEN julianday(p.payment_date) - julianday(fp.first_purchase_date) BETWEEN 0 AND 7 THEN p.amount ELSE 0 END) AS ltv_7_days,
>     SUM(CASE WHEN julianday(p.payment_date) - julianday(fp.first_purchase_date) BETWEEN 0 AND 30 THEN p.amount ELSE 0 END) AS ltv_30_days,
>     SUM(p.amount) AS total_ltv
> FROM customer c
> JOIN payment p ON c.customer_id = p.customer_id
> JOIN (
>     SELECT
>         c.customer_id,
>         MIN(p.payment_date) AS first_purchase_date
>     FROM customer c
>     JOIN payment p ON c.customer_id = p.customer_id
>     GROUP BY c.customer_id
> ) fp ON c.customer_id = fp.customer_id
> GROUP BY c.customer_id
> ```
>
> **Final SQL statement:**
>
> ```sql
> SELECT
>     AVG(ltv_7_days / total_ltv * 100) AS avg_ltv_7_days,
>     AVG(ltv_30_days / total_ltv * 100) AS avg_ltv_30_days,
>     AVG(total_ltv) AS avg_total_ltv
> FROM (
>     SELECT
>         c.customer_id,
>         SUM(CASE WHEN julianday(p.payment_date) - julianday(fp.first_purchase_date) BETWEEN 0 AND 7 THEN p.amount ELSE 0 END) AS ltv_7_days,
>         SUM(CASE WHEN julianday(p.payment_date) - julianday(fp.first_purchase_date) BETWEEN 0 AND 30 THEN p.amount ELSE 0 END) AS ltv_30_days,
>         SUM(p.amount) AS total_ltv
>     FROM customer c
>     JOIN payment p ON c.customer_id = p.customer_id
>     JOIN (
>         SELECT
>             c.customer_id,
>             MIN(p.payment_date) AS first_purchase_date
>         FROM customer c
>         JOIN payment p ON c.customer_id = p.customer_id
>         GROUP BY c.customer_id
>     ) fp ON c.customer_id = fp.customer_id
>     GROUP BY c.customer_id
> ) subquery
> WHERE total_ltv > 0;
> ```
>
> **Case analysis:**
> This is a two-layer nested complex SQL query involving multiple logical operations and nested structures. Compared with the CoT reasoning paradigm, the SQL component pre-generation paradigm simultaneously decomposes the task step-by-step and generates corresponding SQL statements, reducing the complexity of the task. For example, in step one, the model identifies the need to `Calculate the total lifetime sales (LTV) for each customer`, which corresponds to step 3 in the CoT reasoning paradigm, and concurrently produces the corresponding sub-SQL. This reduces the likelihood of errors in the final SQL generation for that sub-task.
>
> ---
>
> **This detailed example demonstrates how our method effectively handles extremely complex SQL queries as found in Spider 2.0 benchmark cases.**

---

### Author Response · Authors · 2025-12-03
**Summary of Rebuttal and Manuscript Revisions**

We would like to thank all the reviewers for their valuable comments and suggestions, which have greatly helped us improve our work. We have carefully addressed all the reviewers’ questions and concerns with detailed explanations and responses, and have completed the corresponding revisions to the manuscript.

Due to the comprehensive nature of our replies, we present below a concise summary of our responses and the main manuscript revision records.

## Rebuttal Summary

**Response to Reviewer 1:**
1. Clarified misunderstandings: Our method achieves better generalization and robustness compared to baselines, while consuming fewer resources.
2. Supplemented experimental results and case analysis on Spider 2.0, demonstrating the effectiveness of our method on more complex SQL queries.
3. Clarified misunderstandings: In real-world scenarios where gold SQL is unavailable, using a trained model is not affected.
4. Added comparison with Alpha-SQL, showing that our method obtains comparable execution accuracy with much lower token consumption and experimental cost, and achieves better execution accuracy on `challenging` data subsets.

**Response to Reviewer 2:**
1. Clarified misunderstandings about contributions: Explained that the SQL components pre-generation method differs in purpose and form from Chain-of-Thought (CoT). Compared with CoT, our method effectively utilizes the information of pre-generated components when producing the final SQL, and experiments validate its effectiveness. Additionally, we innovatively designed a maximum connected subtree reward mechanism based on SQL syntax trees to improve the accuracy of local components.
2. Clarified misunderstandings: Our paper includes experiments where the model is trained on the simpler Spider dataset and validated on the relatively more complex BIRD dataset, illustrating that our method’s effectiveness is not solely dependent on the complexity of the BIRD dataset.
3. Clarified misunderstandings: Supplemented efficiency analysis experiments to show that our method’s efficiency is comparable to the standard GRPO algorithm.
4. Supplemented error analysis of the complexity analysis module, further demonstrating the effectiveness of SQL component pre-generation for complex SQL queries.

**Response to Reviewer 3:**
1. Discussed the potential issue of redundant components generated by the pre-generation scheme, and proposed improvements including AST matching-based penalty mechanisms and reward clipping to address redundancy.
2. Discussed possible impacts of misclassification errors in the complexity analysis module.
3. Discussed limitations of the AST matching reward mechanism, which may mistakenly penalize some semantically correct SQL clauses. Our method mitigates semantic diversity through syntax tree normalization and finer-grained reward matching; experiments show that positive samples receive significantly higher reward values than negative samples.
4. Clarified misunderstandings: Our method employs single-round reinforcement learning (RL) rather than multi-round RL, so training and inference latency are comparable to standard GRPO algorithms; efficiency analysis experiments were supplemented.

**Response to  Reviewer 4:**
1. Carefully checked and corrected typographical and formatting errors in the paper, and provided more detailed descriptions of relevant datasets.
2. Supplemented the performance of the SQL components pre-generation method on extremely complex SQL queries. Case analyses further demonstrate that for extremely complex cases such as multi-level nested SQL, our method can perform component pre-generation layer by layer.
3. Clarified misunderstandings:
  - Our method does not assign zero component pre-generation reward to simple queries, but rather uses a different reward scheme for them.
  - The reward for component pre-generation does not increase continuously with the number of generated components; instead, it is capped by a min function to limit the maximum reward and avoid generating redundant components.


## Manuscript Revision Records

1. Added `Error Analysis of the Complexity Analysis Module` in `Table 7` and `Table 8`.
2. Added analysis of training and inference latency, as well as token consumption, in `Table 9`.
3. Expanded the dataset introduction section in `Appendix A.1`.
4. Added experiments on `Spider 2.0` in Appendix `A4.2`, and case analysis of `Spider 2.0` in `Appendix A4.3`.
5. Added multi-layer nested SQL case analysis to verify whether the model can perform SQL component pre-generation layer by layer, in `Appendix A4.4`.
6. Expanded the introduction and experimental comparison of the differences between the SQL component pre-generation inference paradigm and the CoT inference paradigm in `Appendix A4.5`.
7. Expanded the discussion of limitations and future work in `Appendix B`.

---

### Note · Authors · 2026-01-26

I have read and agree with the venue's withdrawal policy on behalf of myself and my co-authors.

---

### Meta-Review · Area_Chair_7PnY · 2026-01-05

**Summary:**

The paper proposes a method to address "semantic blind spots" in Text-to-SQL generation by pre-generating SQL components before the final query construction, guided by a complexity analysis module and trained via GRPO with an AST-based reward. While the reviewers appreciated the intuitive motivation and the rigor of the ablation studies, the consensus leans towards rejection due to concerns regarding the significance of the contribution and empirical gains. The primary concerns driving this decision are: (1) Marginal Empirical Improvements: As noted by Reviewer jJZB, the method often yields only marginal gains (and sometimes underperforms) compared to strong baselines like Reward-SQL or even standard CoT on certain splits, raising questions about the cost-benefit ratio of the complex pipeline. (2) Limited Novelty: Reviewer mVHC pointed out that the distinction between "SQL component pre-generation" and existing Chain-of-Thought (CoT) or decomposition methods is subtle. While the authors argue that generating SQL fragments bridges a semantic gap, it can be viewed as an incremental variation of established reasoning paradigms rather than a fundamental breakthrough. (3) Pipeline Fragility: Both Reviewers TWiP and mVHC highlighted the modest F1 scores (0.7-0.8) of the complexity analysis module. Since the entire pre-generation strategy hinges on this classification, the error propagation risk remains a significant unresolved weakness.

**Reviewer Concerns:**

Addressed Concerns:
* Missing Baselines & Comparisons: The authors successfully addressed Reviewer jJZB’s request for comparisons with Alpha-SQL and Reviewer mVHC/TWiP’s requests for efficiency/latency analysis. The authors demonstrated that their method is more token-efficient than MCTS-based approaches.
* Spider 2.0 Evaluation: In response to Reviewer jJZB, the authors provided supplementary results on the Spider 2.0 dataset.
* Reward Hacking: The authors clarified the reward clipping mechanism and the binding of complexity analysis rewards to address Reviewer TWiP’s concerns about the model generating redundant components to game the system.
* Presentation: Reviewer H8Vy’s concerns regarding typos and formatting were acknowledged and addressed in the revision plan.

Outstanding Concerns:
* Marginal Performance Gains: Reviewer jJZB’s core concern regarding the method's performance relative to baselines remains the strongest argument against acceptance. While the authors showed better generalization on BIRD compared to SQL-R1, the overall dominance over existing SFT and RL methods is not decisive enough to justify the added complexity.
* Novelty vs. CoT: Reviewer mVHC’s concern regarding the conceptual similarity to CoT persists. While the authors provided a formalization distinguishing the two, the practical difference—generating SQL snippets vs. natural language plans—is viewed as an incremental engineering step rather than a transformative methodology suitable for ICLR.
* Reliability of Complexity Analysis: The error analysis provided in the rebuttal confirmed Reviewer mVHC and TWiP’s suspicions that the complexity classifier is a weak link. The fact that misclassification leads to significant performance drops (as shown in the author's Table 2 in the rebuttal) remains a valid structural critique of the proposed pipeline.

**Reviewer Scores:**

* Reviewer jJZB (Score: 4): This reviewer would likely maintain their score of 4. While the authors added the requested Spider 2.0 and Alpha-SQL data, the results on Spider 2.0 (single-digit to low double-digit accuracy) and the "marginal improvements" argument likely leave the reviewer unconvinced about the method's efficacy.
* Reviewer mVHC (Score: 4): This reviewer would likely maintain their score of 4. The rebuttal clarified the mechanics but confirmed the fragility of the complexity module (the "weak link"). The conceptual overlap with CoT likely prevents them from seeing this as a significant enough contribution.
* Reviewer TWiP (Score: 6): This reviewer explicitly stated "the current score is appropriate" after the rebuttal. They would likely keep their score at 6, acknowledging the method is sound but perhaps noting the limitations of the complexity module.
* Reviewer H8Vy (Score: 6): This reviewer was satisfied with the clarifications regarding pre-generation rewards and multi-level nesting. They would likely maintain a score of 6.

---

### Decision · Program_Chairs · 2026-01-26

Reject